# Applying Super-Resolution and Tomography Concepts to Identify Receptive Field Subunits in the Retina

**Steffen Krüppel**[1,2,3], **Mohammad H. Khani**[1,2,¤a], **Helene M. Schreyer**[1,2,¤a], **Shashwat Sridhar**[1,2], **Varsha Ramakrishna**[1,2,4], **Sören J. Zapp**[1,2], **Matthias Mietsch**[5,6], **Dimokratis Karamanlis**[1,2,¤b], **Tim Gollisch**[1,2,3,7] *

**1** University Medical Center Göttingen, Department of Ophthalmology, Göttingen, Germany, **2** Bernstein Center for Computational Neuroscience Göttingen, Göttingen, Germany, **3** Cluster of Excellence "Multiscale Bioimaging: from Molecular Machines to Networks of Excitable Cells" (MBExC), University of Göttingen, Göttingen, Germany, **4** International Max Planck Research School for Neurosciences, Göttingen, Germany, **5** German Primate Center, Laboratory Animal Science Unit, Göttingen, Germany, **6** German Center for Cardiovascular Research, Partner Site Göttingen, Göttingen, Germany, **7** Else Kröner Fresenius Center for Optogenetic Therapies, University Medical Center Göttingen, Göttingen, Germany

¤a Current address: Institute of Molecular and Clinical Ophthalmology Basel, Basel, Switzerland
¤b Current address: University of Geneva, Department of Basic Neurosciences, Geneva, Switzerland
* tim.gollisch@med.uni-goettingen.de

**Data Availability Statement:** The marmoset spike trains analyzed here have been made available on G-Node (https://doi.org/10.12751/g-node.m4bvl9) together with the code used for analysis. The code

## Abstract

Spatially nonlinear stimulus integration by retinal ganglion cells lies at the heart of various computations performed by the retina. It arises from the nonlinear transmission of signals that ganglion cells receive from bipolar cells, which thereby constitute functional subunits within a ganglion cell's receptive field. Inferring these subunits from recorded ganglion cell activity promises a new avenue for studying the functional architecture of the retina. This calls for efficient methods, which leave sufficient experimental time to leverage the acquired knowledge for further investigating identified subunits. Here, we combine concepts from super-resolution microscopy and computed tomography and introduce super-resolved tomographic reconstruction (STR) as a technique to efficiently stimulate and locate receptive field subunits. Simulations demonstrate that this approach can reliably identify subunits across a wide range of model variations, and application in recordings of primate parasol ganglion cells validates the experimental feasibility. STR can potentially reveal comprehensive subunit layouts within only a few tens of minutes of recording time, making it ideal for online analysis and closed-loop investigations of receptive field substructure in retina recordings.

## Author summary

Neural computations in sensory systems often involve nonlinear pooling of sensory information. In the vertebrate retina, nonlinear signal transmission between bipolar cells and downstream ganglion cells, the output neurons of the retina, shapes the ganglion cells' functional properties and structures a ganglion cell's receptive field into smaller subunits.

used for simulating ganglion cell subunit models has been made available on GitHub (https://github.com/gollischlab/Super-Resolved_Tomographic_Reconstruction).

**Funding:** TG was supported by the European Research Council (ERC) under the European Union's Horizon 2020 research and innovation program (grant agreement number 724822) and by the Deutsche Forschungsgemeinschaft (DFG, German Research Foundation) – project IDs 515774656; 432680300 (SFB 1456, project B05); 390729940 (Germany's Excellence Strategy–EXC 2067/1). SK was supported by the Göttingen Graduate Center for Neurosciences, Biophysics, and Molecular Biosciences at the Georg-August-Universität Göttingen. The funders had no role in study design, data collection and analysis, decision to publish, or preparation of the manuscript.

**Competing interests:** The authors have declared that no competing interests exist.

Methods for identifying these subunits from recordings of ganglion cell activity are needed to better understand the signal flow and the computations occurring between bipolar and ganglion cells. We here show that concepts from super-resolution microscopy and tomography can be used to design a visual stimulus and corresponding analysis to efficiently trigger ganglion cell activity while maintaining high spatial resolution for revealing subunits. As demonstrated by computer simulations and recordings from the primate retina, the method can identify subunit layouts with little experimental recording time, providing for the possibility to be combined with in-depth functional analyses or applied in closed-loop experiments.

## Introduction

Retinal ganglion cells, the output neurons of the retina, are classically modelled with a linear-nonlinear (LN) model [1]. This can take the center-surround structure of their receptive fields into account, but indiscriminately considers luminance signals inside the receptive field to be integrated linearly and passed through a nonlinearity only afterwards, at the model's output stage. While the LN model is still popular due to its simplicity, it has long been known that many ganglion cells respond strongly to spatially structured stimuli, even when there is no net-change in the average illumination of the receptive field [2–6]–a characteristic the LN model cannot replicate. This spatial nonlinearity is mediated via functional subunits in the receptive fields of retinal ganglion cells. These enable various specific computations that would be impossible without them, from sensitivity to fine spatial structures to various types of motion and pattern sensitivity [7–13]. Moreover, nonlinear spatial integration also plays a major role in shaping ganglion cell responses to natural stimuli [14–16]. The biological correlate of receptive field subunits are the retina's bipolar cells, which have been found to rectify their excitatory inputs to ganglion cells [6,17,18].

Given the importance of spatially nonlinear processing in the retina, understanding the underlying circuits is highly desirable. For retinal ganglion cells, extracellular recordings with multielectrode arrays [19] allow large-scale functional characterizations that can capture the diversity of cell types [20–23]. For bipolar cells, progress has been made on large-scale recordings using glutamate imaging [24,25], but it is difficult to combine these techniques to obtain information about the connectivity between large populations of bipolar and ganglion cells. As an alternative, several approaches have been developed to infer subunits and thus connected bipolar cells from ganglion cell recordings alone [26–31]. However, these approaches are often based on ganglion cell responses to fine spatiotemporal white noise stimuli, which generally evoke comparatively weak responses, making long recordings necessary. Limited recording time may thus obstruct reliable subunit identification or curtail the functional investigations of subunit layouts.

To address this issue, we introduce a novel method to identify the layout of subunits composing a ganglion cell's receptive field that makes use of stimuli specifically targeted towards the computational characteristics of subunits and that has the potential of considerably reducing the required recording time. The method combines the functional principles underlying stimulated emission depletion (STED) microscopy [32,33] and tomographic imaging such as computed tomography (CT) scans [34], and we accordingly term it super-resolved tomographic reconstruction (STR). In this manuscript, we investigate the potential of STR via comprehensive modelling and electrophysiological recordings from primate retinal ganglion cells.

## Results

### Super-resolved tomographic reconstruction approach

Retinal ganglion cells often display spatially nonlinear integration of luminance signals [2]. Fig 1A and 1B exemplifies this with responses of an On parasol cell recorded in the isolated marmoset retina. The cell responded strongly to increases of luminance during a full-field stimulus, but not to decreases of luminance (Fig 1A). On the other hand, a stimulus that involved reversals of a spatial pattern while keeping the mean luminance inside the receptive field constant also led to substantial responses (Fig 1B). This nonlinear spatial integration of luminance inside the receptive field of a ganglion cell is mediated via so-called subunits, which are believed to tile the receptive field and to correspond to bipolar cells, the source of the excitatory input to ganglion cells [3,5,6,17,27].

To develop a method for identifying the layout of subunits by recording ganglion cell responses to visual stimuli, we set up computational subunit models to simulate the response to a flashed spatial pattern. For concreteness, we focused the models on On-type ganglion cells. We modelled the nonlinear spatial integration by approximating both the subunits, i.e., bipolar cells, and the ganglion cell itself as separate LN integration stages, yielding an LNLN-like model. The first stage consisted of a linear spatial integration at the level of the subunits, which we portrayed as 2D Gaussian filters applied to the image's Weber contrast values, followed by a rectification. At the level of the ganglion cell, the subunit outputs were then summed before a final transformation resulted in the model's response, given as the average spike count or firing rate elicited by the flashed spatial pattern.

A simple approach to probe the spatial sensitivity profile of a ganglion cell is to flash a small spot of light (Fig 1C, left) at different locations across the cell's receptive field. As illustrated for a schematic model with four circular subunits (Fig 1C, center), the responses of the model (Fig 1C, right) together map out its receptive field, which corresponds to the union of the subunits. Since the subunits have no gaps between each other and even overlap, individual subunits can usually not be identified by reproducing the receptive field.

To enable detection of the subunit structure inside a receptive field, our approach is to add a suppressive dark ring, or annulus, around the excitatory spot, leading to a shape that is commonly known as a Mexican hat (Fig 1D, left). By applying this stimulus, we make use of the linear luminance integration within subunits and the nonlinear integration across subunits. For instance, the average luminance of the hat is the same as the background gray (similar to the example of Fig 1B), but the model still produces responses due to its spatially nonlinear computation (Fig 1D, right). If the size of the hat is similar to the size of the subunits, an individual subunit will only be activated by the stimulus if the hat is placed close to its center. In this case, the suppressive ring of the stimulus plays a subordinate role for that subunit, due to the subunit's greater sensitivity closer to its center, which is a reasonable assumption even if the exact Gaussian shape is a simplification. On the other hand, if the stimulus is placed at the overlap of two or more subunits, each subunit will be triggered significantly less or not at all, because the suppressive ring now strikes the more sensitive central parts of the subunits, while the excitatory white spot only hits the periphery of each of the subunits.

Comparing the response maps obtained with the simple homogeneous spot (Fig 1C) and with the Mexican-hat shaped spot (Fig 1D), similar response strengths are obtained for spots placed at the center of a subunit, because a suppressive ring does not decisively affect that subunit and the suppression of neighboring subunits evoked by the ring is rectified anyway. By contrast, for spots placed at the overlap of subunits, the suppressive ring diminishes responses, as explained above, whereas a simple spot still leads to a strong response because the partially activated subunits do not receive any suppression and combine their activation to accumulate

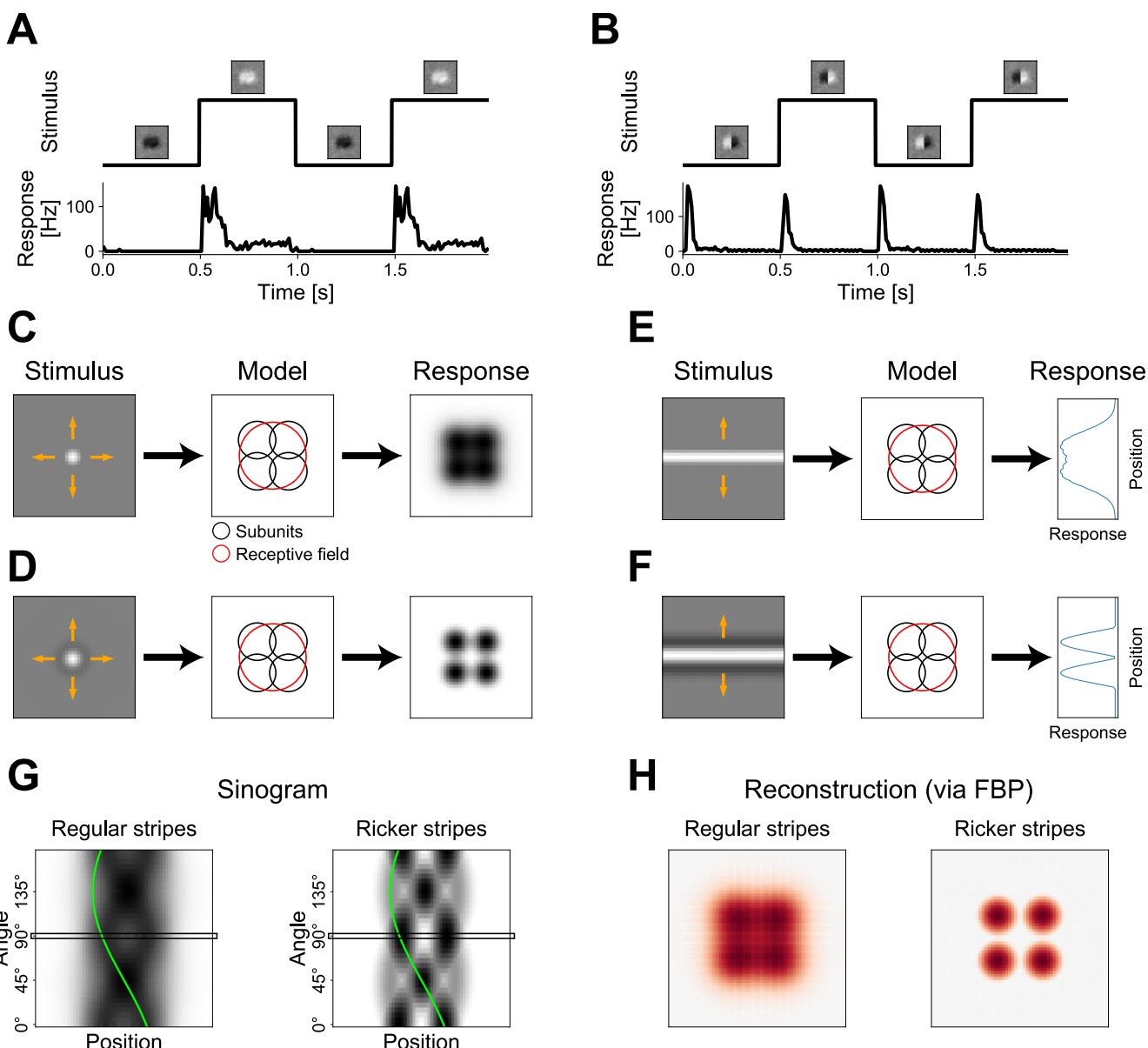

**Fig 1. Schematic of super-resolved tomographic reconstruction (STR) method.** (A) Response of a sample On parasol retinal ganglion cell to light-intensity steps without spatial structure. Top: Stimulus time course with the insets displaying a point-wise multiplication of the cell's receptive field with the current stimulus. Bottom: Peri-stimulus time histogram (PSTH) of the cell's response. For visualization purposes, the PSTH is repeated once. (B) Same cell as in (A), but for a grating stimulus with spatial structure. (C) Schematic depiction of using a spot stimulus (left) to probe the receptive field of a model On-type ganglion cell (center). Orange arrows on top of stimulus signify the shifts of the stimulus for probing the receptive field. Black circles represent the 1.5 σ ellipses of the subunits, red circle represents the 1.5 σ ellipse of the receptive field. Response of the model (right) shows which spot positions led to a strong response (black) and which to a weak response (white). (D) Same as (C), but for a stimulus with an added dark ring around the white spot. (E) 1D probing of model responses with a horizontal stripe (left) at different vertical positions in the receptive field of the same model. The response (right) depended on the vertical position of the stripe. (F) Same as (E), but for a Ricker stripe, which has added dark sidebands adjacent to the white center stripe. (G) Sinograms of the responses of the model to the plain stripes (left) and to the Ricker stripes (right) as measured from 36 stripe angles (steps of 5°) and 60 stripe positions (steps of 2/3 pixels). Dark shading denotes stronger responses. Black rectangles at 90° mark measurements shown in (E) and (F). Green line indicates sine trace of one subunit in the model's layout. (H) Reconstructions of the sinograms in (G) using filtered back-projection (FBP). Red denotes positive values in the reconstruction, blue negative ones.

a strong integrated response. Thus, the suppressive ring effectively leads to a spatial sharpening of the subunits, such that they can be identified more clearly as hotspots in the model's response.

Yet, probing a receptive field with such hat stimuli would be inefficient for two reasons. Firstly, responses of a ganglion cell to an individual hat stimulus would likely be relatively weak and might not even reach spiking threshold, since only a small portion of the receptive field is activated. Secondly, the receptive field would have to be scanned point-by-point over two spatial dimensions requiring a large number of presentations at different locations.

We thus extended the idea of the spatial sharpening of subunit responses by turning towards a tomographic version of the concept. If an extensive white stripe is flashed inside the receptive field (Fig 1E, left), the response of the model will reflect the combined receptive field intensity lying within the confines of the stripe. Accordingly, if multiple positions (here 60) in the receptive field are probed by such a flashed stripe, the responses will represent a projection of the receptive field along the stripe's orientation (Fig 1E, right). Similarly, if the profile of the stripe utilizes the same shape introduced before with an excitatory center band and suppressive sidebands (Fig 1F, left), the responses will represent a projection of the sharpened subunits (Fig 1F, right). We term such a stimulus a Ricker stripe, since we apply a profile that is given by the Ricker wavelet (see Methods).

While individual subunits often cannot be identified in one such projection (in the example, both response peaks are caused by two subunits each), multiple projections can be measured by varying the angle of the Ricker stripe (here using 36 angles in steps of 5°). The resulting data can be displayed in a so-called sinogram, in which each row corresponds to one projection at a fixed angle (Fig 1G). In the sinogram, every subunit leaves a sine-like trace that overlaps with the traces of other subunits for some but not all angles (green line in Fig 1G highlights the trace of a sample subunit), and their traces can thus be disentangled.

To reconstruct the subunit layout from a set of projections compiled in a sinogram, we used filtered back-projection (FBP), one of the most common techniques in the field of tomography [35,36]. The FBP reconstruction of measurements with plain white stripes (Fig 1H, left) resembles the model's receptive field that can also be measured by probing with a plain white spot (Fig 1C). Meanwhile, the reconstruction of measurements with Ricker stripes reveals the locations of the subunits as hotspots (Fig 1H, right) similar to probing with a hat stimulus (Fig 1D). This tomographic presentation, however, has the advantage of evoking stronger responses since subunits are triggered more often and partially simultaneously, thereby reducing experiment time. We will refer to this technique as super-resolved tomographic reconstruction (STR): While the center-surround structure of the Ricker stripes sharpens subunit responses and thereby effectively super-resolves them below the scale of the subunits themselves, the presentation of stripes at varying positions and angles allows for a tomographic reconstruction of subunits.

## Assessment of subunit identification with simulated data

We extended the overly simplified ganglion cell model used in Fig 1 to test STR in a more realistic setting. Firstly, we adopted a procedure to randomly generate layouts of larger numbers of subunits and also took elliptical subunits with varying degrees of overlap into account. Fig 2 shows sample layouts for 6, 10, and 14 subunits (first column). Just as before, the subunits cannot be identified from the structure of the receptive fields alone, even if noise-free high-resolution measurements are considered (Fig 2, second column). The sinograms gathered via the tomographic presentation of Ricker stripes (Fig 2, third column) show a more complex structure than in the previous, simple example of Fig 1, and individual subunit traces are

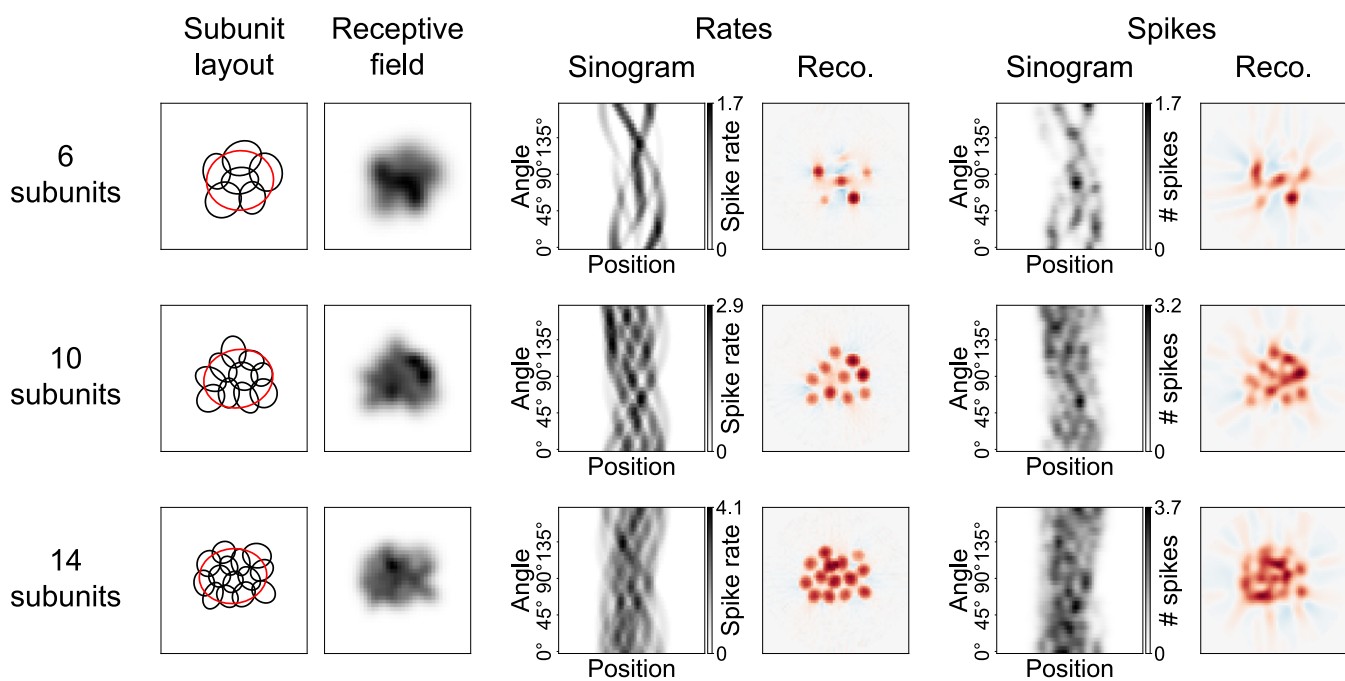

**Fig 2. Application of STR to model simulations with realistic settings.** Three sample layouts with 6 (top row), 10 (middle row), and 14 (bottom row) subunits are depicted. First column shows the subunit layouts, with black ellipses portraying the 1.5 σ ellipses of the subunits and red ellipses the 1.5 σ ellipses of the receptive fields. Second column is the receptive field. Third column contains the sinograms for spike rates, i.e., expected spike counts. Fourth column shows the reconstructions from the sinograms in the third column. Red denotes positive values, blue negative values. Fifth column holds the sinograms for measurements of stochastic spike counts. Each combination of the 36 stripe angles and 60 stripe positions was measured only once. Gaussian smoothing has been applied to these sinograms (described in more detail in the main text). Last column pictures reconstructions from the sinograms in the penultimate column.

progressively harder to distinguish as the model comprises more subunits. Nevertheless, the corresponding reconstructions (Fig 2, fourth column) display a clear hotspot structure with each hotspot representing the location of a subunit.

Next, we extended the model to include the stochasticity of spiking responses by applying a Poisson process for spike generation in order to reflect the inherently noisy responses of real retinal ganglion cells. The Poisson process was used for simplicity, even though ganglion cell spiking is typically more regular than Poisson noise would suggest, so that this can be viewed as a conservative assessment of the effect of spiking variability. To tune the range of obtained spike counts (and thereby the noise level in the simulation), we defined how the expected spike count from the Poisson process relates to a given activation of the model. To do so, we first assumed that a full-field flash of 100% Weber contrast (i.e., "white") would elicit an average response of 30 spikes, whereas no stimulation yielded zero spikes. For any given summed subunit signal in response to flashing a Ricker stripe, we then obtained the expected number of spikes by linear interpolation.

The sinograms in the third column in Fig 2 thus depict the expected spike count (i.e., spike rate in terms of spikes per stimulus), while the penultimate column contains the actual stochastically simulated spike count in response to one flash of the Ricker stripe for each angle and position. Although the addition of stochasticity to the model visibly affects the quality of the sinograms, the resulting reconstructions (Fig 2, last column) still feature apparent hotspot structures, although there is not always a clear one-to-one correspondence of hotspots and simulated subunits. Yet, many subunits can nonetheless easily be identified, demonstrating the potential of STR. Note that the effect of stochasticity could be reduced and reconstructions

improved by averaging over multiple presentations of the same Ricker stripe, which would, on the other hand, correspond to longer recording times.

The more realistic model settings introduced above enable us to make a more meaningful assessment of the influence of certain stimulation and analysis parameters. For instance, to obtain the results presented in Fig 2, we made the following adjustments that we will also use as a default for the rest of this manuscript: We chose the width $w$ of the Ricker stripe profile, defined as the distance of the two zero-crossings, i.e., the width of the white central stripe, to be $w$ = 5 pixels. For comparison, in the case of models with 10 subunits, the average effective subunit diameter (diameter of a circle with the same area as the subunit ellipse at 1.5 $\sigma$ of the Gaussian profile) lies at 7 pixels, and the average effective receptive field diameter at just under 17 pixels. We also strengthened the sidebands of the Ricker stripes by a multiplicative surround factor $s$ = 2.5 to increase their effect of sharpening the subunits. This entails that a Ricker stripe produces a net darkening, since its integral is equal to zero only if $s$ = 1. Finally, sinograms obtained from models employing a spiking process were smoothed by a Gaussian filter across positions and angles with standard deviations of $\sigma_{pos}$ = 2.5% of the simulation area size (1 pixel or 1.5 stripe positions) and $\sigma_{ang}$ = 5˚ (1 angle step), respectively, to alleviate the influence of noise (hence the non-integer grayscale values in the penultimate column of Fig 2).

In order to assess the influence of any of these changes, we attempted to quantify the quality of a reconstruction in terms of how truthfully it reflects the subunit layout. Fig 3A illustrates the subunit layout (1.5 $\sigma$ ellipses) and receptive field of a sample model cell that is analyzed throughout Fig 3, and Fig 3B shows the FBP reconstruction as obtained from stochastic spikes. As noted before, a successful reconstruction is characterized by the hotspots coinciding with the locations of the subunits. We therefore first located the hotspots in the reconstruction by finding all local maxima larger than 30% of the global maximum (Fig 3B, yellow markers). We discarded any hotspots that might have fallen outside a circle with a diameter of 90% of the reconstructed area (Fig 3B, blue circle) to avoid artifacts, which often occurred at the edge of the reconstructed image. This procedure is deliberately simplistic since our aim here is not to present a state-of-the-art detection algorithm for hotspots, but to merely provide a means for a quantitative analysis of the reconstruction.

After finding the hotspots, we determined which of them fell inside the 0.75 $\sigma$ ellipse of a subunit (Fig 3B, black ellipses; double hits counted as one hit and one miss) and calculated the F-score, which is a combined measure of precision and sensitivity (see Methods). The F-score ranges from 0 to 1, with values close to 0 indicating that few subunits had been detected via hotspots (low sensitivity) and/or that there were many hotspots that did not correspond to subunits (low precision), and 1 indicating that hotspots and subunits matched perfectly. In the example of Fig 3B, nine of the ten detected hotspots lay in the 0.75 $\sigma$ ellipse of a subunit, and one subunit was not detected, leading to an F-score of 0.9. Since the noise of the spike generation is an integral part of the challenge of choosing good stimulus and analysis parameters, all considerations made below, including the calculation of F-scores, assume models with a stochastic spike generation process as in Fig 3B.

With this evaluation system in place, we first examined how the quality of the reconstruction depends on the characteristics of the Ricker stripes, i.e., their width $w$ and sideband strength $s$, while fixing the sinogram smoothing parameters $\sigma_{pos}$ and $\sigma_{ang}$ at the default. As expected, by increasing the width of the stripes and decreasing the strength of the suppressive sidebands, the model produced a larger number of spikes in response (Fig 3C, left). While this means that the sinogram is less noisy, it also entails that the reconstruction is more representative of the receptive field instead of the individual subunits. On the other hand, a narrow stripe with strongly suppressive sidebands, supposed to sharpen the subunit responses substantially, results in responses too weak for FBP to reveal many subunits (Fig 3C, right). We thus probed

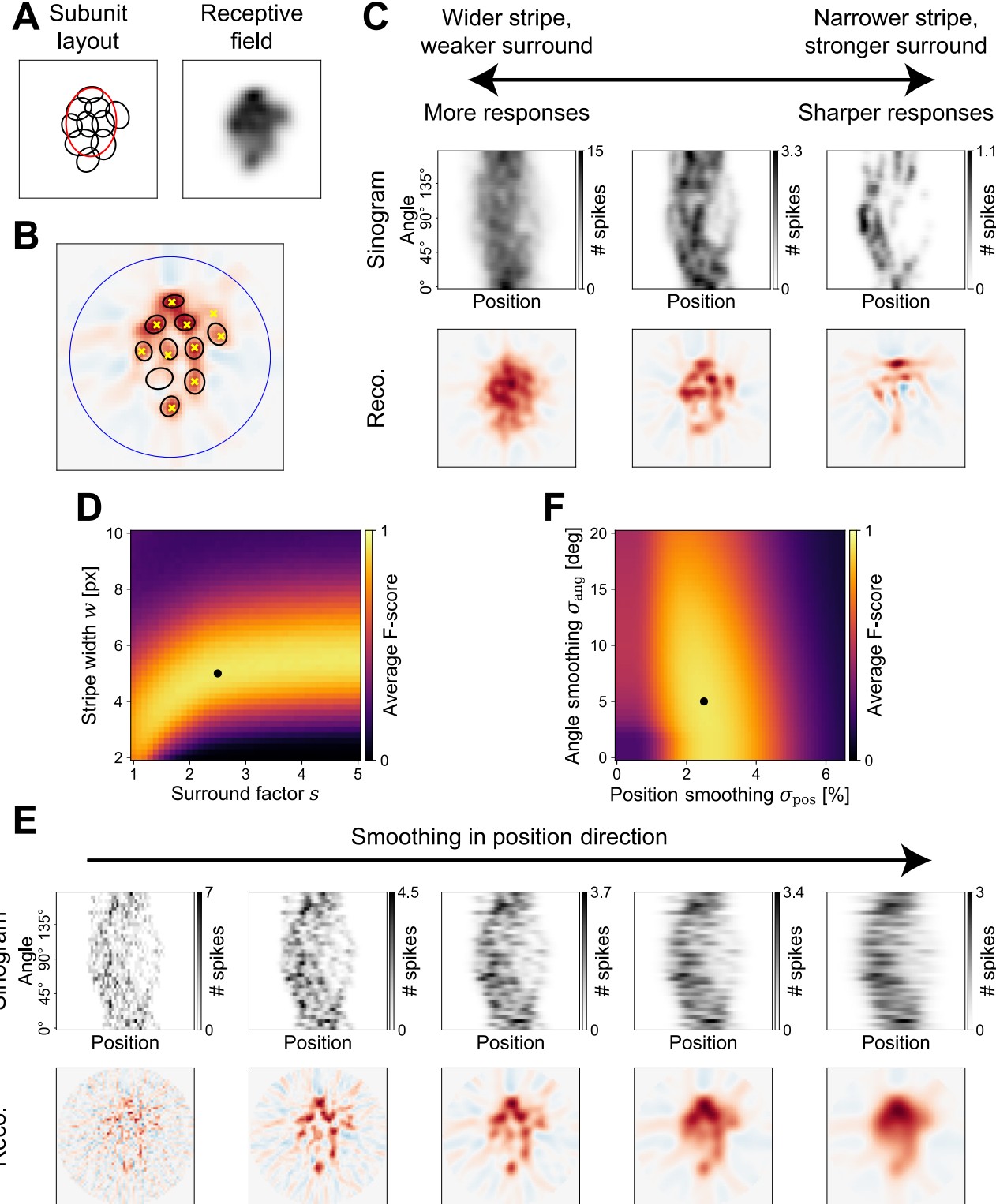

**Fig 3. Optimal stimulus and analysis parameters.** (A) Sample model layout (left) and receptive field (right) used throughout this figure (layout outlines 1.5 σ ellipses of subunits and receptive field). (B) Illustration of the detection of hotspots in a reconstruction. Background image is FBP reconstruction with red and blue colors representing positive and negative values, respectively. Large dark-blue circle depicts area in which local maxima (yellow crosses) are identified. Local maxima are compared with 0.75 σ ellipses of the underlying subunits (black) to compute an F-score. (C) Sample sinograms (top row) and corresponding reconstructions (bottom row) of measurements with varying stimulus parameters. Surround factors s are 1, 2, and 5 from left to right,

stripe width values $w$ are 10, 5, 4. (D) Search for the optimal parameters in the parameter space of surround factor $s$ and stripe width $w$. Brighter colors denote better average F-score for 1000 model instantiations with ten subunits each. Optimal parameters ($s = 2.5$, $w = 5$ pixels) are marked by a black dot. (E) Influence of smoothing the sinogram in position direction on a sample sinogram (top row) and the corresponding reconstructions (bottom row). Standard deviations $\sigma_{pos}$ of the Gaussian filters are (from left to right) 0%, 1.5%, 3%, 4.5%, and 6% of the simulation area size. Smoothing in angle-direction is omitted for these plots ($\sigma_{ang} = 0°$). (F) Like (D), but for search for optimal smoothing of the sinogram in the parameter space of standard deviations for stripe position smoothing $\sigma_{pos}$ (optimum is 2.5%) and stripe angle smoothing $\sigma_{ang}$ (optimum is 5°).

the parameter space of Ricker stripe width $w$ and surround factor $s$ to discover if a proper balance can be found. Fig 3D shows the average F-score STR achieved for layouts of ten subunits, and, indeed, certain combinations of the two parameters led to respectable F-scores. The best F-score of 0.93 was reached using the previously introduced combination of the stripe width $w = 5$ pixels and the surround factor $s = 2.5$ (Fig 3D, black dot).

Similarly, we investigated the effectiveness of smoothing the sinogram in order to overcome the issue of noise. Fig 3E exemplifies the influence of smoothing in the stripe position-direction (without any smoothing across angles) and demonstrates that the noise introduced with the spike generation process greatly impairs the FBP reconstruction if not counteracted (leftmost example involves no smoothing at all). On the other end of the spectrum, too much smoothing naturally eliminates all finer structures and the reconstruction simply reflects the receptive field. Again, a balance must be found and we computed the average F-score of layouts with ten subunits to identify this balance (Fig 3F). Doing so, we determined the optimal standard deviation for Gaussian smoothing to be $\sigma_{pos} = 2.5\%$ of the simulation area size and $\sigma_{ang} = 5°$–the values that we adopted as our default.

With these optimal parameters, STR achieved an average F-score of 0.93 for layouts of ten subunits. The remaining 0.07 in performance were primarily due to undetected subunits (54% of errors), followed by spurious detections (31%). Mislocalized subunits, which we considered to be a hotspot in the 1.5 $\sigma$ but not 0.75 $\sigma$ ellipse of the subunit, only accounted for 14% of mistakes.

## Influence of number of subunits and measurement time

While we here determined the optimal stimulus and analysis parameters for layouts of ten subunits, these parameters also worked well for more or fewer subunits. Fig 4A shows that, in the range of roughly five to fifteen subunits, subunit detection was similarly effective with the default parameters (blue line) as with parameters optimized for each specific number of subunits (red line), and only deviated strongly from the optimal performance for numbers of subunits that were considerably larger or smaller than our standard scenario of ten subunits.

In such cases, it is helpful to adjust the stimulus and analysis parameters. While new optimal parameters could again be identified by a similar analysis as done above for layouts of ten subunits (Fig 3), we found that the optimal parameters largely followed an intuitive scaling behavior reflecting the spatial scale of subunits. In our simulations, subunit layouts were scaled to a constant receptive field size for every number of subunits $N$, so that the diameter of a subunit decreased according to $1/\sqrt{N}$. Since this provides the relevant spatial scaling of the subunit detection task, parameters related to the spatial scale could thus be deduced from the optimal parameters for ten subunits. Specifically, we scaled the stripe width $w$ and the smoothing in position-direction $\sigma_{pos}$ with $1/\sqrt{N}$, while leaving the surround factor $s$ and the angle-smoothing $\sigma_{ang}$ constant (Fig 4B and 4C). We confirmed that, in our simulations, these scaled parameters were indeed optimal–or very close to optimal–by performing analyses similar to the ones in Fig 3D and 3F, and we therefore generally used this scaling relationship to derive the optimal parameters for a given number of subunits. Nevertheless, even with the optimal

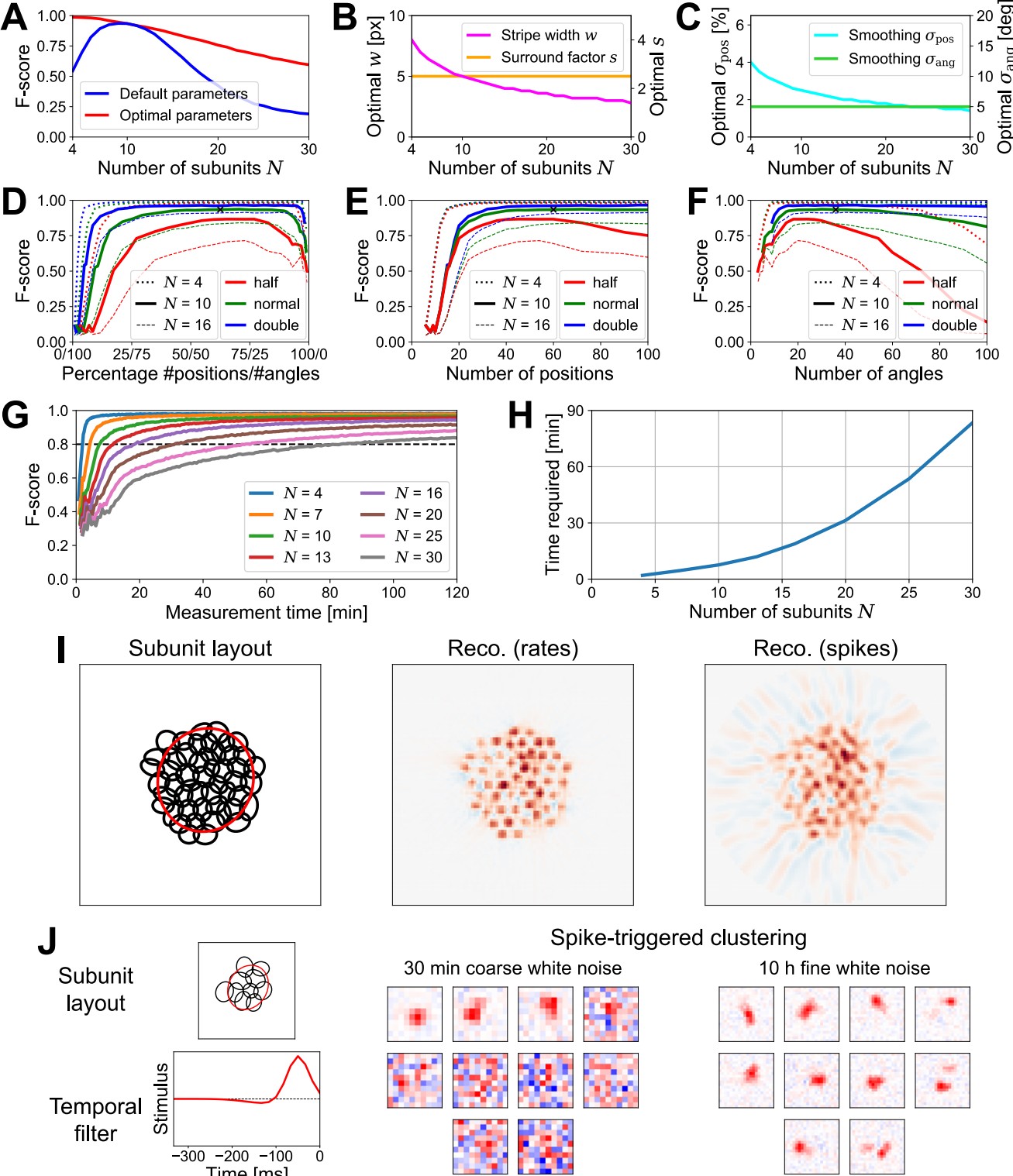

**Fig 4. Measurement time.** (A) Average F-score (over 1000 instantiations) versus number of subunits for the default parameters (blue, optimal for ten subunits) and for parameters adjusted for each number of subunits (red). (B) Optimal stripe width $w$ and optimal surround factor $s$, depending on number of subunits. (C) Optimal position smoothing $\sigma_{pos}$ and optimal angle smoothing $\sigma_{ang}$. $w$ and $\sigma_{pos}$ in (B) and (C) were obtained by scaling with the number of subunits (rounded to an accuracy of 0.2 and 0.1, respectively), and $s$ and $\sigma_{ang}$ were kept constant. (D) Subunit detection performance (F-score) depending on the ratio of the number of stripe positions versus angles for fixed total numbers of stripe presentations (green: 2160, default; red: half of the default; blue:

double the default) and for different numbers of subunits $N$, using corresponding optimal parameters. Black cross marks the default scenario of ten subunits, 60 stripe positions, and 36 stripe angles. (E) Same as (D) but with the absolute number of stripe positions on the x-axis, up to a maximum of 100 positions. (F) Same as (D) but with the absolute number of stripe angles on the x-axis, up to 100 angles. (G) Subunit detection performance depending on measurement time for different numbers of subunits $N$, each with optimal parameters. Dashed horizontal line marks a threshold of 0.8. (H) Measurement time required to pass the 0.8 threshold in (G) depending on the number of subunits. (I) Exemplary subunit layout (left) consisting of 50 subunits, reconstructed with noise-free rate responses (center) and spike responses (right). Ricker stripes were presented at 89 positions (required minimum number according to scaling described in main text) and with 202 angles, equating to 3 hours of stimulation. Stripe and analysis parameters were scaled as before, but with a reduced angle smoothing of $\sigma_{ang} = 2°$. (J) Left: Sample subunit layout with an added temporal filter to simulate responses to spatiotemporal stimuli. Center: Spatial filters from spike-triggered clustering with locally normalized L1 regularization of the sample layout's responses to 30 minutes of coarse binary white noise. Stimulus pixels had a size of 4x4 simulated screen pixels. Red pixels denote positive values, blue pixels negative values, with each filter normalized to its absolute maximum. Right: Spike-triggered clustering with simulated responses to ten hours of fine white noise (2x2 screen pixels).

parameters, layouts with many subunits were harder to uncover than layouts with fewer subunits (Fig 4A, red line). This is not surprising, as more and smaller subunits should require more data with higher spatial resolution, for example, by using more stripe positions at denser sampling, which, in turn, amounts to longer required recording times.

To investigate this issue in more detail and assess the influence of the measurement time, we varied the number of stripe positions and angles. By default, we had tested 2160 stripe presentations, corresponding to all combinations of 60 positions and 36 angles (Fig 4D, black cross). However, using 36 positions and 60 angles instead, led to a similar F-score for our standard simulation condition, as did a variety of other ratios of the number of positions versus angles with the same number of total combinations (solid green line). So long as a certain minimum number of positions as well as angles was tested, the performance plateaued and was indifferent to the ratio of positions versus angles. Similar plateaus were also apparent for a larger (blue lines) or smaller (red lines) total number of stripe presentations, and more (dashed lines) or fewer (dotted lines) subunits.

To better determine the number of positions required to reach the performance plateau, we plotted the same data against the absolute number of positions instead of the ratio (Fig 4E). This demonstrates that the required number of positions was independent of the available measurement time (all dotted, solid, and dashed lines, respectively, rise at the same point, irrespective of color). However, the number and thus size of subunits did influence the minimum number of stripe positions that needed to be tested; smaller subunits required a larger number of stripe positions.

We hypothesize that this reflects a required effective resolution, which can be thought of as the number of stripe positions inside a standard subunit size necessary to distinguish between subunits. For layouts of ten subunits (Fig 4E, solid lines), using 40 positions ensures that the effective resolution does not limit the achievable performance. Given the effective subunit diameter of 7 pixels (derived from the 1.5 $\sigma$ ellipse) in these simulations and the simulation area's size of 40 by 40 pixels, this equates to a required effective resolution of 7 stripe positions inside a subunit. Scaled by $\sqrt{N}$ to layouts of $N = 4$ and $N = 16$ subunits, one would consequently expect to require 25 and 51 total stripe positions, respectively, to reach the same spatial resolution per subunit, both reasonable values according to Fig 4E (dotted and dashed lines). Similarly, a minimum value can also be determined for the required number of angles, but here neither the measurement time nor the number of subunits had a clear impact (Fig 4F).

To investigate how measurement time impacts the performance of STR depending on the number of subunits $N$, we set the number of stripe positions according to the $\sqrt{N}$-scaling derived from the effective resolution as discussed above (using 40 positions for ten subunits) and used the number of stripe angles as a proxy for the measurement time. To convert the total number of stripe presentations into a measurement time, we assumed that each stripe presentation, including a recovery time at homogeneous background illumination, takes 0.6

seconds. The F-score of reconstructions naturally improves with longer measurements, but there is a distinct difference between the quick success with layouts of few subunits and the slow improvements observed with large numbers of subunits (Fig 4G). We then used an average F-score of 0.8 as a threshold (dashed line) to define how much measurement time STR requires for a given number of subunits to be successful. This revealed that the required measurement time grew approximately with the square of the number of subunits (Fig 4H). The reason for this superlinear increase is that the required measurement time is prolonged not only by a larger necessary number of stripe positions to achieve the same effective resolution, but also by, e.g., weaker responses to the narrower stripes needed for layouts with more and smaller subunits. Yet, despite the $N^2$-scaling, layouts with several tens of subunits should still be in reach with reasonable experimental time because of the low baseline of less than ten minutes for identifying layouts with ten subunits.

In principle, there is no fundamental upper limit to the number of subunits that STR can reveal. Even layouts consisting of 50 subunits (Fig 4I, left) could be reconstructed if recording time could be sufficiently extended to average out noise (Fig 4I, center). With three hours of simulated stimulation, which is still in the realm of possibility, the particular sample layout in Fig 4I was reconstructed with an F-score of 0.94, given optimal stimulation parameters (right).

These analyses demonstrate that STR has the potential to rapidly and accurately infer subunit layouts and thereby improve subunit detection over existing methods. To provide a concrete comparison, we used our simulated subunit model and applied spike-triggered clustering, which has been shown to infer subunits from less than 30 minutes of recorded data [31]. Since spike-triggered clustering relies on a spatiotemporal white noise stimulus, we added a temporal filter to our model to be able to simulate the temporal dynamics of such a stimulus (Fig 4J, left). This temporal filter is applied at the subunit stage together with the subunits' spatial filters, thus converting the spatiotemporal stimulus into a sequence of subunit activations. The rest of the model remained identical to our default model, i.e., subunit signals were summed, rectified, and converted into spike counts via a Poisson process. To keep the overall sensitivity comparable, we normalized the temporal filter such that a brief light flash of 150 ms produced the same total spike count as in our default model without temporal dynamics, and we applied a temporal discretization at 60 Hz.

To apply spike-triggered clustering, we stimulated the model with 30 minutes of coarse binary white noise, which led to ~30.000 spikes in response (17 Hz average firing rate). To then extract spatial filters, we supplied the algorithm with the known temporal filter of the model to appropriately filter out the temporal dimension and with the known number of subunits. Based on this data, we indeed managed to infer localized filters with spike-triggered clustering, but these were few and only corresponded to aggregates of the underlying model subunits (Fig 4J, center), in line with the observations in the original presentation of the method [31]. To check whether it was the limited data that prevented spike-triggered clustering from uncovering the true subunit layout, we also used ten hours of finely structured white-noise stimulation, evoking ~300.000 spikes in response (9 Hz). Here, spike triggered-clustering identified many more subunits that matched the true model subunits (Fig 4J, right). Thus, spike-triggered clustering can yield coarse subunits with tens of minutes of data, but seems to require at least an order of magnitude longer for finding the true model subunits as compared to our tomographic approach. This is not surprising given the generic nature of the required spatiotemporal white noise stimulus and the relatively weak activation of subunits, and is a property that spike-triggered clustering shares with other subunit identification methods that work with white noise [27–29].

## Robustness of method

So far, the considered subunit models were all based on the same general structure with fixed characteristics, like the subunit nonlinearity and the shape and overlap of subunits, and only varied in the specific subunit layouts. However, different ganglion cell types have different functional properties, which can also vary depending on, e.g., retinal location [37,38], and many specifics are not known a priori. Identifying subunits with STR, however, is robust against variations in subunit signaling, as revealed by changing various model components.

For example, we increased the overlap of the subunits leading to receptive fields with even less discernible structure (Fig 5, top row, left plots show sample layout and corresponding receptive field). Nevertheless, noise-free sinograms comprised of firing-rate responses appear not visibly inferior to those from the default model composition, and reconstructions calculated from these sinograms still clearly represent the subunit layout (Fig 5, top, center). When a spiking process is included in the model, sinograms and reconstructions also demonstrate decent quality (Fig 5, top, right), with the average F-score decreasing somewhat to 0.84, as compared to the original model characteristics. Much of this loss can be recovered by increasing the stripe width from $w = 5$ to $w = 5.6$ pixels. This is because the increased overlap in our simulations goes along with a larger size of each subunit compared to previous layouts, which makes the wider stripes more suitable. With this change, an F-score similar to the one for the default scenario can be achieved, which might be unexpected, since an increased overlap should make distinguishing the subunits more difficult. However, enlarging the subunits also increases the size of the 0.75 $\sigma$ ellipses used to determine if a hotspot corresponds to a subunit, thereby leading to a potentially better F-score even if the exact same reconstruction would be obtained. Nevertheless, we conclude that an increased subunit overlap is not per se detrimental to STR.

The exact shape of the subunits is also not a sensitive factor. We replaced the Gaussian subunits of the standard model with subunits whose cross-sections reflect the positive part of a cosine curve between zero-crossings (Fig 5, second row; for comparability, subunit ellipses depict the 1.5 $\sigma$ ellipses of Gaussians fitted to the cosine subunits). Again, the reconstructions clearly exhibit a hotspot structure that reflects the subunit layout and the average F-score increased marginally to 0.94.

Other changes to the model's properties, like modifying the subunit nonlinearities, can have stronger effects. We replaced the default rectified-linear transformation of subunit signals with a rectifying nonlinearity that additionally squared all positive values (Fig 5, third row). Doing so leads to the subunits being directly apparent in the receptive field, because weak excitation of multiple subunits at their overlap evokes a weaker model response than strong excitation of a single subunit at its center owing to the squaring of subunit signals. Note, however, that Fig 5 shows a noise-free high-resolution receptive field measurement that is virtually impossible to achieve in a real experiment, and a more realistic spike-triggered average (STA) under spatiotemporal white noise would likely not reveal any subunits. While the reconstruction of noise-free responses to Ricker stripes still flawlessly represents the subunits, the reconstruction obtained from stochastic spiking responses was certainly undermined by the nonlinearity change, with the average F-score decreasing to 0.76. This performance loss seems to stem from the noticeably weaker responses to the Ricker stripes (cf. the ticks at the sinograms' grayscale bars) and thus inferior signal-to-noise ratio. This, in turn, is a result of our choice to set the reference point for the spike count at the maximum response, that is, the response to a full-field white flash. For Ricker stripe stimuli, which activate the model less strongly than the full-field flash, the quadratic nonlinearity therefore weakens the responses compared to the rectified-linear nonlinearity. Indeed, the F-score can be improved up to 0.88

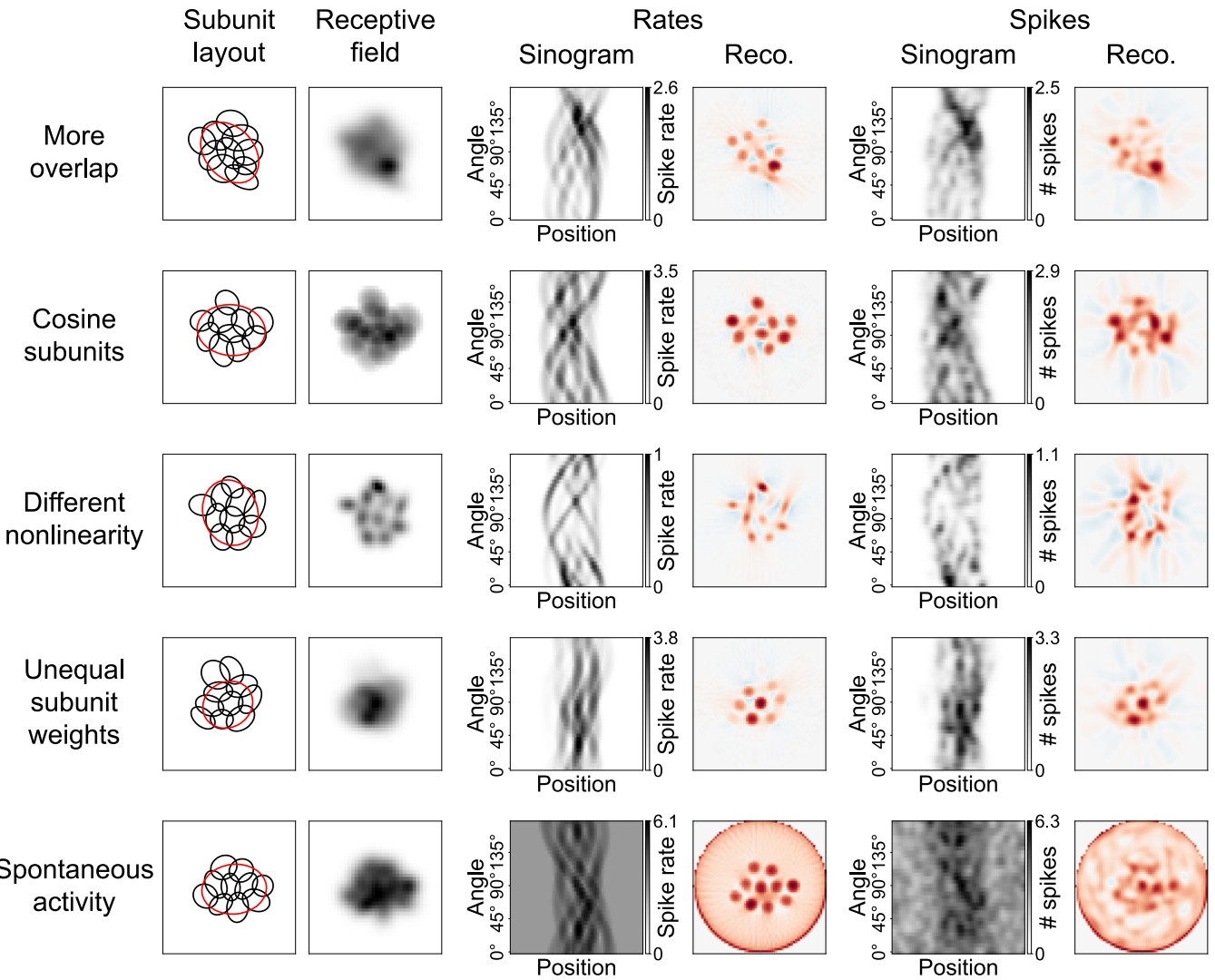

**Fig 5. Robustness of STR to model variations.** Each row demonstrates the effect on STR of one variation of the model via a sample simulation. Layout of the rows is the same as in Fig 2. Top row shows a model with increased subunit overlap (see Methods for details), apparent from the 1.5 σ subunit ellipses. In the second row, the Gaussian-shaped subunits were replaced by subunits with a cosine profile. For comparability, the ellipses in the subunit layout depiction are 1.5 σ ellipses of Gaussians fitted to the cosine subunits. Third row contains a replacement of the rectified-linear nonlinearity with a rectified-quadratic nonlinearity. Weights of the subunits in the fourth row were not all equal as in the default model, but chosen according to a large spatial Gaussian. In this example, the strongest subunit weight was roughly eight times that of the weakest weight. In the bottom row, a base activity of three expected spikes was added to all responses.

by making the Ricker stripes wider from $w = 5$ to $w = 6.2$ pixels, which, as noted in Fig 3, increases the response strength.

In the simulations, we had so far assumed all subunits to contribute with an equal weight to the response of the model, but subunits might realistically contribute differentially, with subunits farther from the cell's center potentially having a weaker connection to it. We modelled this hypothesis by choosing subunit weights according to a 2D spatial Gaussian that prefers subunits close to the center. Consequently, subunits in the periphery of the receptive field played a minor role in activating the modeled ganglion cell, a fact demonstrated by the top two subunits in the sample layout in Fig 5 (fourth row) being almost irrelevant for its receptive field. While more central subunits are still well represented in the reconstructions, these outer

subunits are difficult to recognize, thus reducing the average F-score to 0.76. On the other hand, the significance of the F-score is limited in this case, because it values all subunits equally, which does not reflect their true contribution to the model's responses.

As a final variation of the standard model, we added spontaneous activity to the measurements (Fig 5, bottom row), which we implemented by increasing the expected number of spikes for the Poisson process by 3 irrespective of the stimulus. This is a considerable level of background activity, as it is similar to the maximum systematic response modulation evoked by our stimuli. One effect on the FBP reconstruction is a characteristic artifact ring at the periphery of the reconstructed image. This in itself does not strongly compromise the subunit reconstruction, as is apparent from the high quality of the noise-free reconstruction. Yet, for the automated hotspot detection, we had to confine potential hotspot locations to a circle with a diameter of 90% of the reconstruction region. More importantly, however, the overall noise in the reconstruction is strongly increased by adding spontaneous activity, leading to a decrease in the average F-score to 0.58. The F-score could be improved to 0.70 by making the Ricker stripes slightly wider with $w = 5.2$ pixels and increasing the standard deviations of the Gaussian smoothing of the sinogram from $\sigma_{pos} = 2.5\%$ to $\sigma_{pos} = 3\%$ and from $\sigma_{ang} = 5°$ to $\sigma_{ang} = 7.5°$, thereby counterbalancing the noise to some degree. Yet, substantial levels of background noise can evidently have a significant influence on the quality of the FBP reconstruction.

In addition to uncertainty about model details like the subunit nonlinearity, the retinal circuitry also differs from the simple LNLN structure with a single subunit mosaic that we have assumed here. For example, parasol ganglion cells in the primate retina receive input from multiple types of bipolar cells [39–43], which may form independent superimposed subunit layouts. We approached this issue with models that also received signals from two separate subunit layouts, each contributing 50% of the input weight. The subunits in these layouts might be entirely equivalent and only distributed at different positions (Fig 6A, left, first layout consists of black ellipses, second layout of green ellipses). In this case, the standard stimulus (Fig 6A, center) produced a reconstruction that represented a combination of both layouts (Fig 6A, right). Coinciding subunits were reflected by strong hotspots (left arrow), whereas other hotspots may correspond to a subunit from only one layout (middle arrow), or appeared to merge subunits from differing layouts (right arrow). When combined with the unreliability of spiking, this complicates the interpretation of the reconstructions, even though using the hotspots to predict the subunit locations of each layout still yielded an average F-score of 0.70 with respect to the task of detecting a single of the two layouts.

In most cases, however, one would expect the subunits from different layouts to differ in some characteristics, like their size (Fig 6B). We therefore combined a layout of four large subunits (black ellipses) with another layout of 16 small subunits (green ellipses). Interestingly, varying the stimulus and analysis parameters can now be used to distinguish between the layouts. By applying wide Ricker stripes with $w = 8$ pixels and strong smoothing with $\sigma_{pos} = 4\%$, thus using values optimized for four-subunit layouts, the large-subunit layout is identified quite well (Fig 6B, top row). This approach achieved an average F-score of 0.84 for the large layout, compared to 0.99 for sole four-subunit layouts. With narrow stripes and weak smoothing ($w = 4$ pixels, $\sigma_{pos} = 2\%$), on the other hand, the layout with small subunits could be reconstructed (Fig 6B, bottom row), reaching an average F-score of 0.73, compared to 0.83 for sole 16-subunit layouts. Consequently, size differences between the subunits of superimposed layouts can be exploited by targeting stimulation and analysis to recover both layouts independently.

Incident subunit signals may also differ in their response polarity (Fig 6C), with one layout consisting of regular On subunits (black ellipses) and the other composed of Off subunits

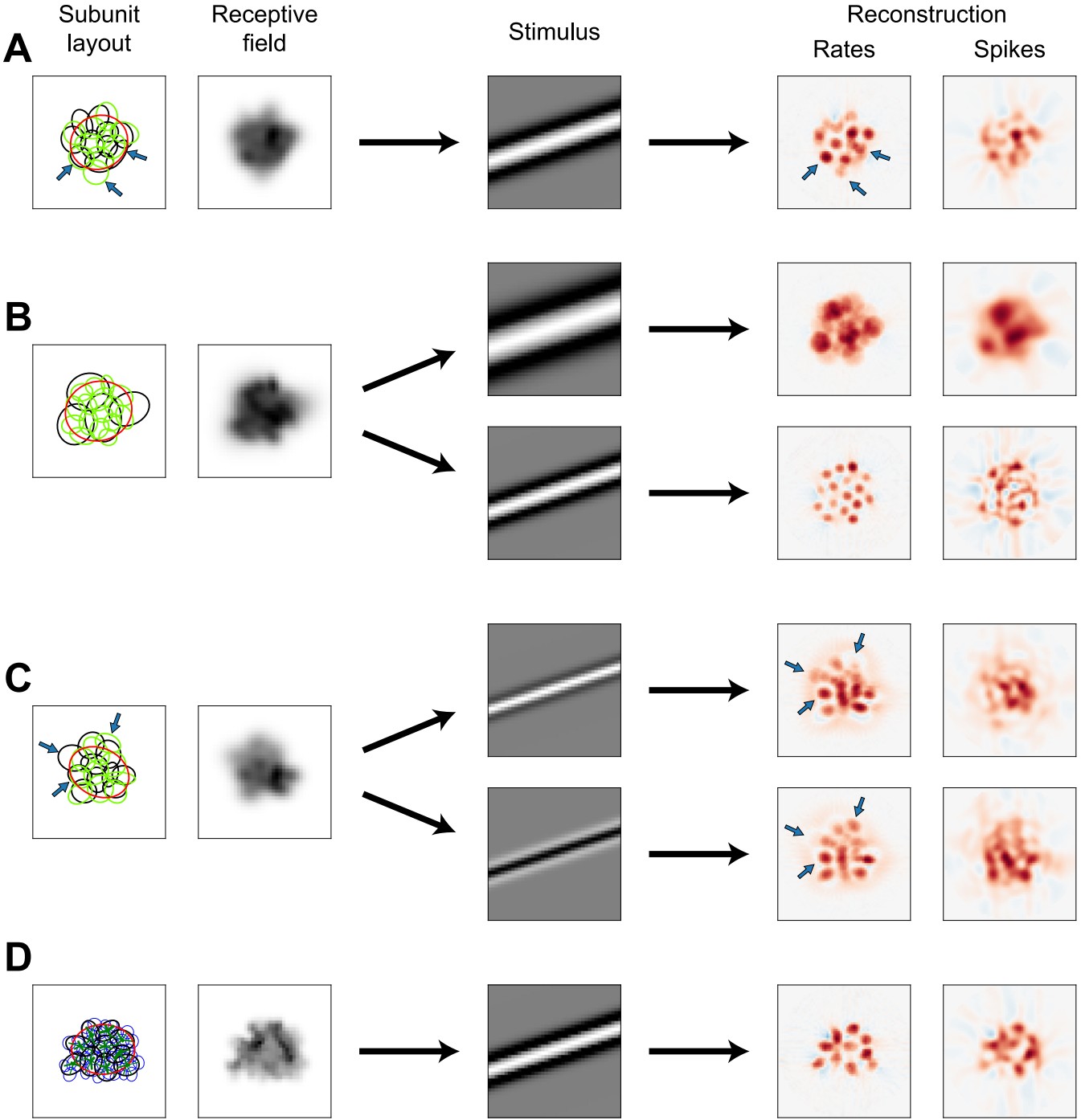

**Fig 6. Variations to the model structure.** (A) Receptive field and subunits (left) of a model with two superimposed subunit layouts (black and green ellipses, respectively), each consisting of ten subunits. Measurements with Ricker stripes (center) yields reconstructions depicted for rate responses and spiking responses (right). Arrows highlight subunits/hotspots described in main text. (B) Same as (A), but for a model with two layouts of differently sized subunits, one containing four subunits (left, black ellipses) and one containing 16 subunits (green ellipses). Measurements with wide Ricker stripes (top row) lead to different reconstructions (right) than measurements with narrow stripes (bottom row). (C) Same as (A), but for a model with one On (black ellipses) and one Off (green ellipses) subunit layout. The depicted receptive field is for On-type stimulation. Measurements with stripes of different polarity (top and bottom row) lead to different reconstructions. (D) Depiction of an LNLNLN model with Gaussian photoreceptors (blue ellipses, representing 1.5 σ contours). Green lines display the connection weights between photoreceptors and subunits by their width.

(green ellipses). This suggests using Ricker stripes of differing polarity to recover each subunit layout independently of the other. However, the strong sidebands of the Ricker stripes pose a potent stimulus for the layout of the opposite polarity, causing strong interference and preventing straightforward independent analysis. This can be mitigated somewhat by drastically lowering the surround factor from $s = 2.5$ to $s = 1.0$ among other changes ($w = 3$ pixels, $\sigma_{ang} = 7.5°$) to reconstruct the On and Off subunit layouts individually (Fig 6C, top and bottom row). Using these settings, coinciding subunits from the On and Off layouts are the most reliable to be reconstructed (bottom left arrow), similar to the case of independent layouts of same size and same response polarity (Fig 6A). In addition, some On subunits are only visible in the On reconstruction (top left arrow) and some Off subunits only in the Off reconstruction (top right arrow). However, due to the necessary changes to the stimulus, hotspots are not well separated and responses weak, such that reconstructions from spiking responses are of medium quality only, reaching an average F-score of 0.67 for the layout of the targeted polarity.

Subunit models as the ones discussed here typically assume that the subunits themselves integrate sensory signals in a linear fashion and the first nonlinearity occurs at their output. For the vertebrate retina, however, nonlinearities prior to the bipolar-cell-mediated subunits have been reported, as seen by spatially nonlinear integration in the bipolar cell membrane potential [44] or in nonlinear signal transformations by photoreceptors [45–47]. We simulated such a scenario by creating a mosaic of photoreceptors (Fig 6D, left, blue ellipses) and connecting these to the bipolar cell subunits (black ellipses, thickness of green lines signifies connection strength; see Methods for details). Photoreceptor signals were nonlinearly transformed by a piecewise linear function with a slope of 0.5 below and unity above zero, reflecting the fact that photoreceptors do not fully rectify their output and that this first nonlinearity is likely much weaker than the subsequent one at the bipolar cell output. The additional nonlinear-integration stage gives rise to an LNLNLN model. For the ability of STR to identify the (bipolar-cell-mediated) subunits, however, moderate nonlinear integration within subunits only had a minor effect. By just slightly adapting the stimulation parameters ($w = 4$ pixels, $s = 3.0$), subunits could be reconstructed almost as effectively as before (Fig 6D, right), reaching an F-score of 0.90 compared to the initial 0.93. Furthermore, one might speculate that the layout of photoreceptors could also be identified if sufficiently fine Ricker stripes are applied, similar to the scenario of Fig 6B. In our simulations, this was indeed the case with noise-free rate responses, but the responses evoked by the narrow stimuli were too weak for reconstruction from spiking responses. In total, the analyses of Figs 5 and 6 demonstrate that STR is robust against many modifications to the structure and details of our ganglion cell model.

## Limitations of reconstruction with filtered back-projection

While FBP is a simple and easy-to-use algorithm to reconstruct the subunit layout, it is not designed for our specific reconstruction problem. In particular, the spatial width of the applied Ricker stripes means that the receptive field structure near the tested position also influences responses. Furthermore, the nonlinearities inherent in the ganglion cell model represent a deviation from the Radon transform for which FBP is designed as a reconstruction method. We therefore considered how the specifics of FBP may limit its suitability for subunit detection. Aside from the susceptibility to noise in the measurements, as discussed above, a particular caveat concerns the reconstruction of elliptical subunits. Indeed, the circular symmetry of subunits that we had assumed in Fig 1 is clearly an abstraction, and elliptical shapes with different levels of eccentricity are rather the norm in the retina [27,31].

For our stimulation with Ricker stripes, this means that the mean luminance inside an elliptical subunit, even if the stripe precisely hits its center, can still depend heavily on the stripe's

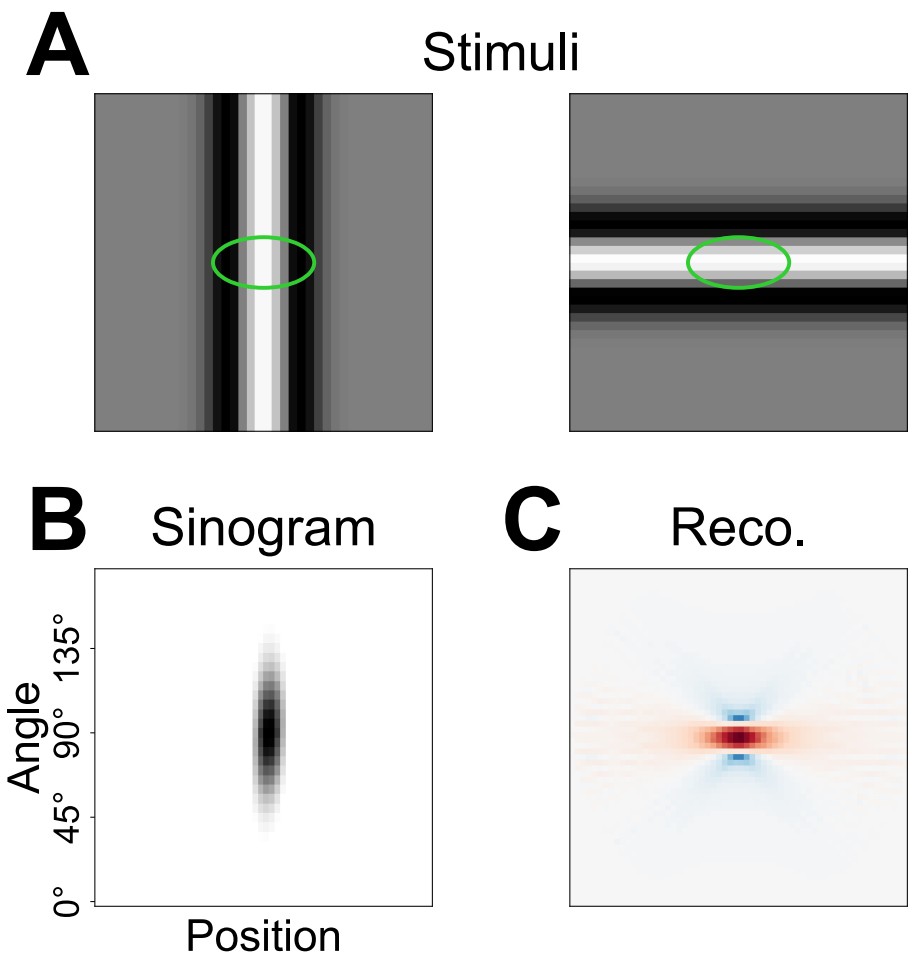

**Fig 7. Shortcomings of FBP as a reconstruction method for subunit layouts.** (A) Sample stimuli of a vertical (left) and a horizontal (right) Ricker stripe hitting the center of an exemplary elliptical subunit (here in green). (B) Rate responses of a model consisting of only that one subunit depicted as a sinogram. (C) Resulting reconstruction via FBP from the sinogram in (B). Red denotes positive values, blue negative values.

orientation. If the stripe is oriented perpendicular to the major axis of the subunit (Fig 7A, left), the suppressive sidebands may influence the subunit much more than for a stripe oriented parallel to it (Fig 7A, right). Consequently, the response elicited by that subunit can depend drastically on the angle of the stripe and might even be completely suppressed for some angles (Fig 7B). In X-ray tomography, this phenomenon would correspond to strongly anisotropic absorption, which FBP does not take into account. Instead, the reconstructed subunit is smeared out along its major axis and negative troughs are reconstructed adjacent to it (Fig 7C, red denotes positive values, blue negative values). While the elliptical subunit can still readily be identified in this simple example consisting of just the one subunit, these effects can overlap in more complex layouts and undermine the quality of the FBP reconstruction. The extent of this effect also depends on the subunit nonlinearity–the rectifying and squaring nonlinearity mentioned before, e.g., will magnify this issue. Nevertheless, as demonstrated before, FBP reconstructions still reproduce the locations of many subunits. Yet, by taking the effects described here into account, an alternative reconstruction method might help locate subunits more consistently, reduce trough effects in the reconstruction, and determine the subunit shape more accurately.

## Experimental test

In order to experimentally test our method, we performed an electrophysiological recording of ganglion cells in an isolated marmoset retina using a multielectrode array while projecting light stimuli onto the retinal photoreceptors. Using responses to a spatiotemporal white noise stimulus, we identified On and Off parasol cells, which we expected to exhibit the most pronounced subunit structure due to their spatially nonlinear characteristics. Both cell types displayed a fast biphasic filter, tiling of visual space by their receptive fields, and consistent autocorrelation functions and output nonlinearities (Fig 8A). Off parasol cells had an average effective receptive field diameter of 111 ± 9 μm (mean ± standard deviation) and On parasol cells of 138 ± 11 μm, both values in line with dendritic field sizes in the marmoset peripheral retina [48].

For the experiments, we adjusted the Ricker stripe stimulus in two major ways: Firstly, we decided to mainly target Off parasol cells, which, for the macaque retina, had been shown to have stronger spatial nonlinearities in the receptive field than their On-type counterparts [15,49]. We therefore flipped the polarity of the Ricker stripes, now using a black center and white sidebands, which are excitatory and suppressive, respectively, for Off cells. Second, to be able to probe a larger portion of the tissue simultaneously, we employed multiple parallel Ricker stripes (Fig 8B shows an excerpt of the screen). The stripes had a safety distance of 375 μm between them to ensure that only one stripe hits a given receptive field center for any stimulus presentation. While a neighboring stripe may fall into the surround of a cell's receptive field, its effect will most likely only be weakly modulating the response because of the distance from the receptive field center and because of the presumably larger spatial scale of nonlinearities in the surround [50]. The Ricker stripes were flashed for 153 ms separated by 447 ms of full-field gray background illumination, and all spikes occurring during the flash of the stripes were used to compose the sinograms. Our choice of stimulus parameters was guided by our simulations with some adjustments. In the simulations, we had used a simulated area of 40 by 40 pixels, with about 20 by 20 pixels being occupied by the receptive field of the model.

In the experiments, the effective receptive field diameters of Off and On parasol cells corresponded to 15 and 18 screen pixels, respectively, such that parameter values are roughly comparable. We decided to use a stripe width of $w = 6$ pixels and a surround factor $s = 1.5$. Reducing the surround factor was motivated by our expectation of incomplete rectification by the subunit nonlinearity, resulting in a component of linear spatial integration, which could weaken ganglion cell responses and decrease the signal-to-noise ratio. Similarly, we chose the slightly larger stripe width to make sure that cells with a supposedly easier-to-reconstruct small number of subunits were stimulated appropriately. The number of stripe positions (75, covering a distance of 50 pixels in steps of 2/3 of a pixel) and the number of stripe angles (36) matched the default choice of the simulations. In the experiments, each combination of position and angle was presented three times to yield an averaged response.

Measured sinograms (Fig 8C) showed a strong overall curvature, even extending beyond the edges of the sinogram, owing to the receptive fields–in contrast to the setting in the simulations–not being centered on the screen. Note that the sinograms here are cyclical, because each stimulus consists of multiple Ricker stripes. That is, when the overall structure in the sinogram crosses the edge, this corresponds to a transition from one specific stripe hitting the receptive field at the farthest shift in one direction, to the neighboring stripe hitting the receptive field at the farthest shift in the other direction. This effect can easily be compensated for, however, by redefining the zero-position in the sinogram. For a given angle, we set the zero position to the stripe position that was closest to the receptive field center known from white noise stimulation. Fig 8D shows such a corrected sinogram, including a Gaussian smoothing

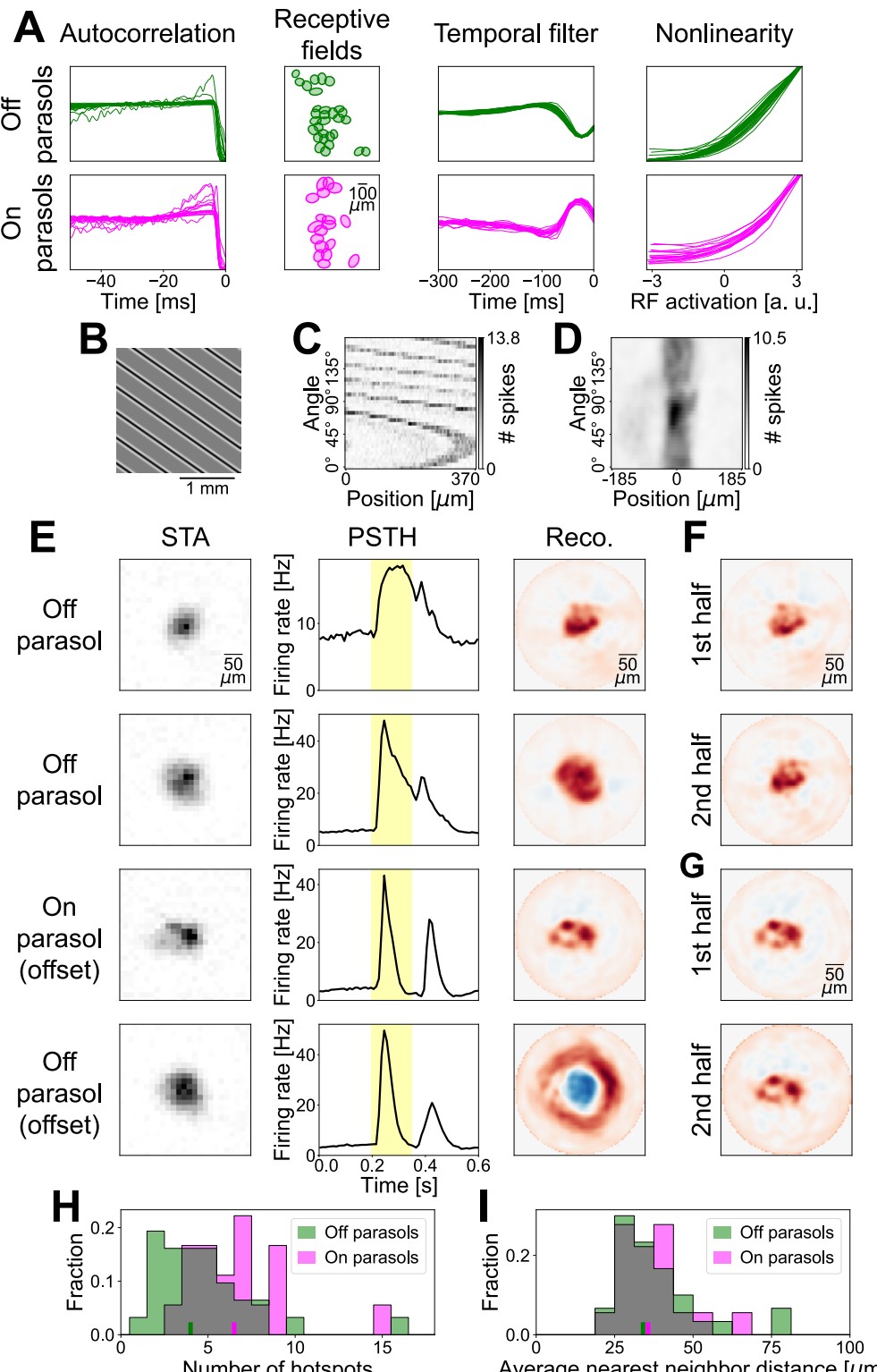

**Fig 8. Experimental application of STR.** (A) Autocorrelation functions, receptive fields (RFs) displayed as 1.5 σ ellipses of Gaussian fits (one distant cell not included), temporal STAs (normalized to unit Euclidean norm), and nonlinearities (scaled to equal maximum) of all identified Off (top) and On (bottom) parasol cells. (B) Excerpt of a sample stimulus projected onto the retina during a flash. (C) Unprocessed sinogram of a sample Off parasol cell. (D) Same sinogram as in (C), but processed by correcting for the receptive field position and applying a Gaussian filter. (E) Overview of the results

of four sample cells. Left column depicts spatial STAs, middle column illustrates PSTHs (yellow background designates presentation of the Ricker stripes), right column shows reconstructions from the processed sinograms via FBP. Red colors in reconstructions denote positive values, blue colors denote negative values. Spatial scales of STAs and reconstructions are equal, but reconstruction has higher resolution. PSTHs were computed irrespective of the angle and position of the stripes with a bin size of 10 ms. Sample cell in top row is same cell as in (C) and (D). Bottom two rows contain the results of an analysis of the offset responses of cells. (F) Reconstructions of the sample Off parasol cell from the top row of (E) from separate analyses of the first and second halves of the measurement. (G) Same as (F), but for the On parasol cell from (E). (H) Distribution of the number of hotspots identified in the reconstructions across all recorded Off and On parasol ganglion cells. Colored ticks at the bottom mark the medians for the two cell types. (I) Distribution of the average distance of a hotspot to its nearest neighbor in each ganglion cell's reconstruction. Ticks at bottom mark the medians.

with a standard deviation of $\sigma_{pos} = 7.5$ µm (which is equivalent to the 2.5% of the simulation area size used before) and $\sigma_{ang} = 5°$ applied. This makes the sinogram similar in appearance to the smoothed sinograms in the simulations analyzed above, which had led to successful detection of the model's subunits.

STAs of Off and On parasol cells computed from an hour-long white noise stimulus with high spatial resolution displayed generally little structure inside the receptive fields (Fig 8E, left column, shows four sample cells). The average peri-stimulus time histograms (PSTHs) for the tomographic stimulus, calculated by averaging over all stripe positions and angles, demonstrate a strong response to the onset of the Ricker stripe flash and also a noticeable response to its offset for most cells (Fig 8E, middle column, yellow background marks duration of stripe flash). The upper two sample reconstructions in Fig 8E (right column) were calculated for two Off parasol cells from the responses that occurred during the stripe flashes. Hotspots are less clearly delineated as in the simulations, but can generally still be recognized.

In addition to the strong response to the onset of the Ricker stripes, the offset response can also be analyzed. This is most informative for On parasol cells, where an analysis of the offset response via the same methods described above for the onset can also reveal a hotspot structure in the reconstruction that is not apparent from the STA (Fig 8E, third row). The sample On parasol cell in Fig 8E is the best example in our dataset and shows four clearly distinct hotspots suggesting that the receptive field of that cell is composed of four subunits. Consequently, while the stimulus we used in the experiments was targeted towards Off cells, the receptive field substructure of cells of the opposite polarity can still be examined by focusing on the offset response.

Finally, we also inspected the offset response of Off parasol cells. Under the applied dark-center Ricker stripes, where the stimulus offset amounts to a brightening at the stripe center and a darkening of the sidebands, responses are mostly triggered when the sidebands fall onto the receptive field center and the stripe center onto the surround. Consequently, the reconstruction primarily uncovered a center-surround structure, but little substructure either in the center or the surround. Although one might hope that this offset analysis might reveal subunit structure in the surround of Off parasol cells, the stimulus parameters are unlikely to be suited for this. Since the average luminance of the Ricker stripes is dominated by the sidebands (due to the surround factor $s > 1$) and spatial integration in the surround of ganglion cells might occur on different spatial scales than in the center [50], the surrounding ring in the reconstruction presumably reflects activation of the receptive field center by the offset of the suppressive sidebands, rather than a response of the surround itself. Consequently, for a given cell, only the responses that correspond to the stimulus part where the center of the Ricker stripe undergoes a light-intensity step with the cell's preferred contrast (for the dark-centered stripes applied here, these are the onset responses for Off cells and offset responses for On cells) appear to be relevant for our approach.

Whether the hotspot structure in the reconstructions actually corresponds to subunits and potentially bipolar cells or is just a result of the noise of the measurement is not immediately clear, as we do not have a ground truth available for the experimental data. Thus, to probe the reliability of the recovered substructure in the receptive field, we split the measurement of the tomographic stimulus into a first and second half and analyzed them separately for comparison. The resulting reconstructions demonstrate that similar receptive field substructure may result from independent data sections (Fig 8F, cell shown is the top Off parasol cell from Fig 8E, and Fig 8G, sample cell is the On parasol cell from Fig 8E), corroborating the biological origin of the derived structures. This was not as apparent for all cells in our dataset, however, as other sample cells could show misaligned hotspots from the two halves of the data or unclear structure. This could suggest that the reconstructions were governed by noise and that a reliable subunit structure may not be discernible with STR in these cells. On the other hand, responses of these cells might not have been sufficiently stable over the recording duration, or using only half of the data, where some stimuli were presented only once, others twice, could be insufficient to get reliable reconstructions.

Despite the remaining uncertainty regarding the experimentally obtained reconstructions, we aimed at exploring their characteristics across the population of recorded cells. We therefore applied the same hotspot detection used previously, identifying local maxima surpassing 30% of the global maximum, and interpreted the hotspots as centers of identified subunits. In this analysis, the number of hotspots found for Off and On parasol cells differed systematically (Fig 8H, Mann-Whitney U test: $p = 0.007$). While Off parasol cells had a median of four hotspots, the median for On parasol cells lay at 6.5. These numbers are roughly in line with previous functional studies [31,51], but lower than expected from an anatomical viewpoint [9,39], potentially because not all subunits, e.g., ones with weaker connections to the ganglion cell, could be detected.

We also aimed at extracting the size of putative subunits, even though the reconstruction is aimed at providing their locations, but not their shape or outline. Yet, the size of subunits should be similar to their distance to their neighbors, since the subunits can be expected to tile visual space. We therefore calculated the average nearest neighbor distance of hotspots in each ganglion cell's reconstruction. This yielded similar sizes for Off and On subunits (Fig 8I, Mann-Whitney U test: $p = 0.87$), with medians of the average nearest neighbor distances at 34 μm and 36 μm for Off and On parasol cells, respectively. These values are consistent with dendritic tree sizes of diffuse bipolar cells in the marmoset retina (diameters typically ranging from 30 μm up to 80 μm [41,42,52]), which are thought to represent the primary source of excitatory input to parasol cells. Thus, the analysis of the recorded data indicates the feasibility of the approach for real ganglion cells, but more systematic experimental explorations will be required in the future to assess the biological insight that can be gained from this approach to subunit identification.

## Discussion

Spatially nonlinear integration of luminance signals inside the receptive fields of ganglion cells is mediated via subunits, which are thought to correspond to the retina's bipolar cells [17]. Inferring the subunit layout from electrophysiological measurements of a ganglion cell promises a new avenue towards understanding the functional properties of the retina's circuitry. Several studies have proposed methods for subunit inference [26–31], but these typically require long recordings with finely structured stimuli that are inefficient in driving responses of ganglion cells. This makes it difficult, for example, to retain sufficient experiment time for studying the uncovered structure in more detail and relate it to functional analyses of stimulus

encoding. We here introduced the method of super-resolved tomographic reconstruction (STR) that combines concepts underlying STED microscopy and tomography to identify subunits from retinal ganglion cell recordings. STR consists of flashing Ricker stripes–named after the Ricker wavelet that describes their center-surround profile–with varying angles at different positions in the receptive field (Fig 1).

The evoked ganglion cell responses can be arranged in a sinogram and the subunit layout can be reconstructed from that sinogram, for example by applying filtered back-projection (FBP). Simulations demonstrated that hotspots in the FBP-based reconstruction reliably corresponded to subunits (Fig 2). To optimize performance, especially when data are noisy, stimulus and analysis parameters can be tuned according to the response properties of the investigated neurons (Fig 3). This enables accurate identification of subunits and provides a substantial decrease of required recording time compared with previous methods (Fig 4). STR also proved to be robust against varying the specifics of the subunit model (Fig 5) or deviations from the classical subunit model structure with a single mosaic-like layer of subunits (Fig 6), even though distortions of the reconstructed subunit layout can occur for non-circular subunits when employing FBP as a reconstruction algorithm (Fig 7). Application to recordings of parasol ganglion cells in the primate retina indicated the experimental feasibility of the approach, although further experimental explorations will be required to evaluate the reliability of the results (Fig 8).

## Relation to STED microscopy and tomography

Our super-resolution approach is conceptually related to stimulated emission depletion (STED) microscopy [32,33]. In regular confocal fluorescence microscopy, a specimen containing fluorescent molecules is scanned with an excitatory spot of light. The resolution of this microscopy technique is determined by the size of the area that emits fluorescent light, which, in turn, is given by the size of the excitatory spot of light. Due to Abbe's diffraction limit, however, a spot of light cannot be focused to an arbitrarily small size, thereby limiting the resolution. STED microscopy therefore introduces a second source of light, shaped like a ring, which depletes fluorescence. Any remaining fluorescent light emission is thus confined to areas that were covered by the excitation but not depletion light. Consequently, the area emitting fluorescent light shrinks, thereby improving the resolution compared to confocal fluorescence microscopy beyond Abbe's diffraction limit.

In the context of subunit identification, the spatial extent of the subunits takes the role of Abbe's diffraction limit. Pairs of subunits that overlap are difficult to separate by spots of light alone, just like pairs of fluorescent molecules with a distance less than the diffraction limit cannot be excited separately. However, the addition of the suppressive ring around the spot of light shrinks the area in which subunits are responsive to the stimulus. A subunit not quite centered with regard to the stimulus is excited by the central spot, but simultaneously "depleted", i.e., suppressed, by the ring of opposing contrast, just like the fluorescence signal from an off-center molecule in STED microscopy is suppressed by the depletion ring surrounding the center of the excitation. Like in STED microscopy, this effect here leads to a super-resolution of the subunit layout beyond what simple spot-like stimulation would suggest.

As a second concept, we introduced a tomographic variation of the super-resolution stimulus in order to evoke stronger responses and sample the receptive field more efficiently. Bars of light have long been used to qualitatively study receptive field properties of visual neurons [53,54], and Sun and Bonds [55] introduced the application of filtered back-projection to quantitatively determine the receptive fields of cells in the cat lateral geniculate nucleus (LGN).

Since then, this approach has been employed to locate receptive fields in various systems including the goldfish, zebrafish, and mouse retina [56,57], primate LGN, V1, and V2 [57–59], and even fMRI of human visual cortex [60]. Previous studies of retinal ganglion cells have shown that the tomographic analysis scheme can markedly accelerate receptive field estimations [56,57], compared to stimulation with individual spots of light or spatiotemporal white noise. We believe the same effect to apply in our tomographic approach with super-resolution stimuli for subunit identification, as Ricker stripes constitute a much more potent stimulus for driving ganglion cells than small spots with suppressive rings or spatiotemporal white noise at the required high spatial resolution (Fig 4).

Nevertheless, differences between the ganglion cell system investigated here and X-ray tomography may complicate the analysis of the sinograms and limit the applicability of FBP. The FBP algorithm provides a discrete approximation of the inverse Radon transform [34]. In the context of X-rays, the Radon transform describes the absorption of the beams travelling through an object. In more general terms, it calculates projections of an object in various directions. If the stimulus used here was an infinitesimally narrow bar without suppressive sidebands, the Radon transform would perfectly describe the responses of our standard ganglion cell model. However, since the Ricker stripes have a finite width and the suppressive sidebands trigger the nonlinearities in the system, the Radon transform reflects only an approximate description of the system. Consequently, deviations occur in the inverse Radon transform, e.g., when the nonlinearity of an elliptical subunit is triggered differentially for different angles (Fig 7). Thus, improvements to the reconstruction of the subunit layout could come from replacing or amending FBP by an appropriate reconstruction algorithm that does not rely on the inverse Radon transform. Iterative reconstruction methods or deep learning approaches potentially combined with LNLN models to replace the Radon transform and with appropriate regularization schemes are promising starting points [34,61–64].

## Identification of subunits from reconstructions

The main objective of the present work has been to obtain reconstructions that can be viewed as a representation of the subunit layout within the receptive field of a retinal ganglion cell. In the idealized scenario of simulated, noise-free ganglion cell responses, the individual subunits were easily identifiable as hotspots in the reconstructed image. When noise or other complications were added to the simulations or when experimental data were considered, the correspondence of hotspots and subunits became less clear. For simplicity, we evaluated the reconstructions by detecting hotspots as local maxima, but future applications may take a more sophisticated approach, such as fitting a mixture of Gaussians to the reconstruction or employing advanced blob-detection algorithms [65]. Clearly, however, the success of any subunit identification technique will depend on how distinct and reliable the hotspots in the reconstructed images are.

Regarding our experimental results, the observation of partially differing hotspot locations obtained from the first versus second half of the recording (Fig 8F and 8G) exemplifies the limits of our approach. Multiple causes for the observed differences are conceivable, and most of them could be remedied in future experiments. For example, the recording quality of the analyzed experiment may not have been sufficient. In particular, the ganglion cells measured here might not have maintained sufficiently stable responses during the recording to warrant a comparison of responses separated by about 45 minutes. Electrophysiological measurements from isolated retinas can decrease in quality over time with responses often becoming more sluggish later on, which can have an effect on the cells' response accuracy to the fine Ricker stripes. In this case, the recording's first half (or first trial) might in fact have yielded a good

correspondence of hotspots and subunits, but without knowledge of the ground truth this can hardly be determined. Simultaneous recordings of connected bipolar cells, as previously performed in the salamander retina [27], might help assess whether identified subunits do indeed represent bipolar cell receptive fields, but such experiments have been difficult, in particular in the mammalian retina.

Another potential source of hotspot and subunit discrepancy might lie in our choice of stimulus parameters. We took a conservative approach, opting for somewhat wider Ricker stripes with weaker sidebands than our simulations suggested, in order to ensure adequate response strength. That goal was easily met, but at the same time there appears to be too little sharpening of subunits, judging from the lack of distinctly separated hotspots in experimental compared to simulated reconstructions. This could also be a reason for why we identified significantly lower numbers of hotspots than the expected number of bipolar cells anatomically connected to each ganglion cell [9,39]. On the other hand, functional subunits may differ from anatomical bipolar cells, at least in primate [31,51]. Nevertheless, in future applications, a general line of thought to follow when deciding on the parameters of the Ricker stripes would be to choose their width $w$ to be similar to or slightly smaller than the expected diameter of subunits and increase the sideband strength $s$ as much as the evoked responses permit to sharpen the contributions of individual subunits.

Furthermore, we decided to target our stimulation towards Off parasol cells by using Ricker stripes with dark centers and bright sidebands. This was motivated by previous observations that Off parasol cells receive more strongly rectified input signals than On parasol cells in the macaque retina [15,49]. Here, we observed that offset responses of On parasol cells are not inferior for our analysis to onset responses of Off parasol cells, which is in line with previous findings about spatially nonlinear responses of parasol cells to grating on- and offsets [49]. Alternatively, it could suggest that marmoset On parasol cell measurements are more suitable for our method than expected, in line with recent observations that, in the marmoset retina, On parasol cells may display particularly strong spatial nonlinearities [66].

Other limits of subunit identification from experimental data may also stem from the investigated retinal circuitry itself. While On and Off parasol ganglion cells in the primate retina are often considered to receive most of their excitatory synaptic input from a single type of bipolar cells each, namely diffuse bipolar cells DB4 and DB3a, respectively, other bipolar cell types may also contribute substantially [39–43]. We have shown in Fig 6 that input from two superimposed subunit layouts can make subunit identification more difficult, but not impossible. In addition, we had there assumed both layouts to contribute equally strong inputs. In reality, often one bipolar cell type can be assumed to provide principal, though not exclusive input, which should facilitate reliable reconstruction of the corresponding dominant subunit layout. Bipolar cells are also connected via gap junctions [67], which support, e.g., motion sensitivity [68], and these couplings could influence spatial stimulus integration and effectively blur individual subunits. Moderately nonlinear spatial integration in the receptive fields of subunits themselves, as observed for some bipolar cells in the salamander retina [44], however, did not pose a major issue for STR (Fig 6D).

Additional complications in the experimental data, compared to the simulations, may arise from the receptive field surround of bipolar cells [69,70] and from interactions with inhibitory amacrine cells. The center-surround structure, however, would be in line with the profile of the Ricker stripes and should thus rather add to the sharpening of responses when the stripe is aligned with the center of a subunit. More problematic could be the influence of amacrine cells. A multitude of different amacrine cell types are known [71], but their functions remain largely unclear [72]. One hypothesis is a linearization of the nonlinear bipolar-to-ganglion cell synapses [49,73]. Such effects could hamper the success of STR, since it relies heavily on the

rectification of excitatory subunit signals. Pharmacological intervention to block inhibition could be a way to better isolate the feed-forward signal processing structure investigated here.

## Implications for circuit analysis

Despite the caveats regarding the correspondence of hotspots in the reconstruction of experimental data to subunits, STR has the potential to identify subunits in a time-efficient manner. Our simulations suggest that far less than half an hour of recording time can be sufficient (Fig 4H), and we partly also observed significant hotspot structures in FBPs from relatively short experimental measurements (Fig 8G). The potential to acquire sufficient data with such short measurements is due to the Ricker stripes serving as a more potent stimulus for driving ganglion cell responses than, e.g., fine spatiotemporal white noise. In addition, no computationally intensive post-processing is required, such that accurate reconstructions of subunit locations could be available within less than 30 minutes of experiment time, which is significantly faster than existing methods [26,27,31]. A further reduction might be achieved by optimizing the presentation times of the Ricker stripes or employing more advanced reconstruction techniques to deal with the limited data. Since this could also make reconstructions of layouts with large numbers of subunits feasible (Fig 4I), STR could be useful for resolving the discrepancy between the higher number of subunits expected from anatomical studies [6,9,39] compared to findings in functional studies [26,27,31,51].

Rapid subunit identification leaves ample recording time to make use of the obtained subunit layouts for in-depth studies of the retinal circuitry. By knowing the subunit locations, targeted stimuli can be manufactured to characterize the spatiotemporal integration properties, nonlinearities and functional synaptic weight of each subunit, and evaluate their similarity across multiple subunits within one ganglion cell receptive field. Interactions between bipolar cells, e.g. via gap junctions, underlying computational functions like motion processing [68,74], could also be investigated more closely by stimulating individual subunits in a specific temporal order. Moreover, since ganglion cell receptive fields tile visual space, one can expect two neighboring ganglion cells to share some of their subunits, corresponding to shared excitatory input [27]. By studying the same subunit via responses of multiple ganglion cells, ganglion and bipolar cell effects can be disentangled and processes like global and local contrast adaptation [75–77] be investigated in greater detail.

Furthermore, since the number of ganglion cell types generally exceeds the number of bipolar cell types [20], some, if not all, bipolar cell types provide input to multiple ganglion cell types, so that, consequently, the subunit layouts of some ganglion cell types should coincide. If such common subunit layouts could be identified, this would provide a way of studying how amacrine cells shape the signals that bipolar cells send to ganglion cells [25,78]. In theory, by identifying all subunit layouts, relating subunit properties to bipolar cell types and studying the transmission strength of each bipolar to ganglion cell type by targeted stimuli, a fairly complete functional connectome of this part of the retina might be in reach, at least as far as connections with sufficiently nonlinear transmission are concerned.

## Potential method extensions

One possible adaptation of the Ricker stripes that might be helpful for some ganglion cell types could be the inclusion of color. For example, the small bistratified ganglion cell in the primate retina is characterized by blue On and yellow Off responses, which are likely conveyed by excitatory input from two types of bipolar cells–a blue-sensitive On bipolar cell and a yellow-sensitive Off bipolar cell [79] (but see [80]). Both can be expected to form overlapping subunit layouts, and we have shown in simulations that two layouts of opposing polarity are not

straightforward to resolve (Fig 6C). Nevertheless, by using stimuli with an appropriate chromatic makeup to independently activate different photoreceptor populations [81], the color-specificity of the small bistratified ganglion cell's inputs could be exploited. For example, S-cone-isolating Ricker stripes with an On-type blue center could be applied to reconstruct the blue-sensitive subunits and, conversely, S-cone-silencing stripes with an Off-type center to reconstruct the yellow-sensitive subunits.

Some of the analysis concepts introduced here may also be of interest for studying nonlinear processing beyond the retina. Subunit models have also been applied to primary visual cortex [82–84] and higher motion processing cortical areas [85,86] as well as to the auditory system [87–89]. Complex cells in the primary visual cortex, for example, display strongly nonlinear response characteristics which can be modeled by subunits that resemble simple cells [54,90] in a way comparable to the subunits of nonlinear ganglion cells [91]. A clearer picture of the organization of these subunits may help understand the functional circuitry of the nonlinear computations in complex cells. For this, the development of suitable visual stimuli may be guided by the idea that the spatial pattern should not necessarily maximize the responses of a subunit but rather sharpen its responses to differentiate it from contributions of other subunits. For a complex cell that prefers edges of a particular orientation, for example, such a stimulus might be a localized, sharp black-white edge whose intensity profile rapidly falls off as one moves away from the black-white transition, or it might comprise an edge with sidebands of opposing contrast on both sides, analogous to the center-surround structure of the Ricker stripes. The desired effect would be to make the response of the complex cell strongly sensitive to the actual positioning of the stimulus relative to the underlying subunit to probe whether the presumed subunit model holds and to potentially identify individual subunits.

More generally, an essential design principle that underlies subunit identification with STR is the application of a stimulus structure that can strongly trigger individual subunits but simultaneously restricts the positions in the probed stimulus space at which responses will occur, so that the effective subunit overlap is reduced. This design principle should be transferrable also to other sensory systems, e.g., by designing auditory stimuli, potentially combined with quasi-tomographic presentation in spectro-temporal space to increase response strength of putative subunits, for identification of functional circuitry underlying nonlinear computations.

## Methods

### Ethics statement

All experiments were performed in conformance with national and institutional guidelines and as approved by the institutional animal care committee of the German Primate Center and by the responsible regional government office (Niedersächsisches Landesamt für Verbraucherschutz und Lebensmittelsicherheit, Permit 33.19-42502-04-20/3458).

### Ganglion cell model

In order to study the super-resolved tomographic reconstruction (STR) method in a scenario with known ground truth, we employed an LNLNP model to simulate ganglion cell responses to a flashed presentation of a given stimulus (grayscale image). The model consisted of the following stages: the linear spatial filters that represent the subunits (L), the subunit nonlinearities (N), the weighted linear summation of subunit signals (L), the output nonlinearity (N), and the spike-generating Poisson process (P).

The subunits mark the first linear computation of the stimulus. The simulated visual area was 40 by 40 pixels large and stimulus pixels could attain values from -1 (black) to +1 (white)

with 0 corresponding to mean gray. Each subunit was modeled as a 2D Gaussian, whose parameters were two standard-deviation values, an angle of rotation, and the x- and y-position of the center situated in the simulated 40-by-40 pixel space. All subunits were normalized to a volume of unity. For our standard model, we only used positive values for the subunit filters, corresponding to On-type subunits (and, downstream, to simulated On-type ganglion cells).

The biologically inspired subunit layouts used throughout this manuscript were generated using Voronoi diagrams of perturbed hexagonal lattices. To create such a layout, we first constructed a large hexagonal lattice (i.e., the centers of a honeycomb structure) with a nearest-neighbor distance of 1/8th of the simulated area. The points in the lattice were then randomly and independently perturbed by shifting them in x- and y-direction by distances each drawn from a Gaussian distribution with a standard deviation of 21% of the nearest-neighbor distance. Next, we determined the Voronoi cells of all perturbed points using the Euclidean distance and picked those $N$ cells (with $N$ being the number of desired subunits) whose centers of mass were closest to the center of the lattice. We then fitted 2D Gaussians to these Voronoi cells that we would use as the subunits. The overlap of the subunits was increased by multiplying their standard deviations with 1.35 and the size of the layout was rendered roughly independent of the number of subunits by scaling with $1/\sqrt{N}$ in both spatial directions. In most cases throughout this manuscript, we typically used $N = 10$ subunits. Note that our procedure of fitting (elliptical) Gaussians to Voronoi cells leads to subunits with varying sizes, eccentricities, and orientations.

To calculate the response of the model to a given stimulus, the linear response of each subunit was first computed as a weighted sum of the stimulus pixels, with weights given by the subunit's Gaussian profile. The linear subunit responses were then passed through the subunit nonlinearities, which we modelled as half-wave rectifications. Next, the subunit outputs were summed in a weighted manner, with all weights being equal and normalized to unity sum.

The resulting signal was then transformed into a spike rate by the output nonlinearity, which here amounted to a simple scaling of the signal as rectification was not required due to the already rectified inputs. As reference points, we assumed that background gray (i.e., a signal of zero) would elicit no spikes and that a full-field white flash (i.e., the maximum stimulation) would yield an average response of 30 spikes. Spike rate responses to all other stimuli were determined by linear interpolation between these two reference points. The resulting spike rate was used as the model output in analyses that were based on noiseless responses. By contrast, when stochastic spike counts were analyzed, the resulting spike rate was converted into an actual spike count using a Poisson process. Here, a single random number was drawn from a Poisson distribution with an expected value given by the spike rate. Thus, the response of the model to a given stimulus was either a spike rate (deterministic) or a random integer spike count (stochastic), depending on the specified model analysis.

The receptive field of a model was empirically determined from responses to individually presented white pixels, with the strength of the receptive field given by the rate response to the white pixel at the corresponding location. For rectifying-linear subunit nonlinearities, this noise-free high-resolution measurement corresponds to a (weighted) sum of the Gaussian subunits.

## Variations of the standard ganglion cell model

For the schematic introduction of STR in Fig 1, the realistic subunit layouts computed from Voronoi diagrams were replaced by a simplistic layout of four subunits at x- as well as at y-positions of 3/8 and 5/8 of the extent of the simulated area. These Gaussian subunits had standard deviations of 1/10 of the extent of the simulated area in both directions and thus no orientation.

To simulate responses to spatiotemporal binary white noise required for the comparison with spike-triggered clustering in Fig 4J, we added a temporal filter to the model. Instead of responses to separate stimuli, this enables computing responses to a sequence of stimuli (frames). The temporal filter is applied together with the spatial Gaussian subunit filters to the stimuli to generate a sequence of subunit activations. These are then passed through the subunit nonlinearities etc. to calculate a sequence of model responses. We have assumed a frame rate of 60 Hz and normalized the temporal filter such that a presentation of any stimulus for 9 frames (150 ms) would elicit the same summed response as in our default model without a temporal filter.

In Fig 5, we separately tested five variations of different parts of the standard model. To have more strongly overlapping subunits (first row), we multiplied the standard deviations of the 2D Gaussians fitted to the selected Voronoi cells during subunit creation with 1.6 instead of 1.35. For cosine-shaped subunits (second row), the value of a subunit at a specific distance from its center was taken from a cosine curve up to its first zero-crossing, with zero elsewhere. In this case, the size of the cosine subunit was chosen such that a Gaussian fitted to it had the required standard deviations and rotation angle. As a different subunit nonlinearity (third row), we applied a threshold-quadratic transformation instead of rectification, which involved an additional squaring of positive values at the output of the subunits. To investigate the influence of differential subunit weights (fourth row), we evaluated a 2D Gaussian centered in the simulated area with a standard deviation of 0.12 times the extent of the area at the center positions of the subunits to obtain their weights. Again, weights were normalized to a sum of unity. Finally, to include spontaneous activity (fifth row), we added a universal baseline level of 3 spikes per stimulus to the model's spike rate.

We also investigated the effect of a second superimposed layout (Fig 6A–6C). Both layouts were generated entirely independently, potentially with differing numbers of subunits. To avoid correlated positioning of subunits, each layout was shifted such that a Gaussian fitted to its individual receptive field (the linear combination of its subunits) would be centered in the simulation area, and rotated by a random angle. Both layouts contributed equally to the model's responses, which was achieved by normalizing the sum of each layout's subunit weights to the same value, here 0.5. In the case of Fig 6C, the polarity of the second layout was flipped such that the subunits were negative filters.

To simulate the effect of spatially nonlinear integration within subunits (Fig 6D), we set up an LNLNLP model where the first stage corresponded to photoreceptors. To do so, we first generated a layout of Gaussian subunits and, analogously, a layout of Gaussian photoreceptors. For the photoreceptor layout, we set the diameter to half of that of the subunits, thus targeting a number of photoreceptors approximately four times the number of subunits. After creating an extensive layout of photoreceptor Gaussians of the desired size, covering the entire simulated area, we discarded all photoreceptors with centers outside the 1.5 σ ellipses of all subunits. We then connected each remaining photoreceptor to any subunit whose 1.5 σ ellipse covered the photoreceptor's center point and applied a weight to this connection according to the value of the subunit's Gaussian at the location of the photoreceptor. Weights incident to a subunit were normalized to a sum of unity. The response to a stimulus was then calculated by first computing the photoreceptor activations as a sum of the stimulus weighted with a photoreceptor's Gaussian profile. Next, the activations were passed through a piecewise linear transformation, where negative values were divided by two and positive values left unchanged. These photoreceptor signals were then combined into subunit activations by a weighted sum according to the previously described connection weight between photoreceptors and subunits. The rest of the model remained unchanged.

## Stimulus and analysis in simulations

To demonstrate the effect of a suppressive ring around an excitatory spot when probing the receptive field with the spot (Fig 1C and 1D), we used a stimulus in the shape of a 2D Marr wavelet:

$$L(r) = \left(1 - \frac{2r^2}{d^2}\right) \cdot e^{-\frac{2r^2}{d^2}}$$

Here, $L(r)$ is the stimulus intensity (Weber contrast) of a pixel at a distance $r$ (in pixels) from the center of the wavelet, and $d = 2\sqrt{5}$ defines the size of the wavelet. The wavelet will take on the maximum allowed intensity of +1 in the center and is normalized to an average intensity of zero. When demonstrating the responses of the model to a spot without a ring, we used the same formula but truncated negative values. These stimuli were centered at every pixel of the simulated area to record responses.

The tomographic stimulus is a stripe with the profile being described by a modified 1D Ricker wavelet, hence the name Ricker stripe:

$$L(x) = s_0(x) \cdot \left(1 - \frac{4x^2}{w^2}\right) \cdot e^{-\frac{2x^2}{w^2}}$$

with

$$s_0(x) = \begin{cases} s, |x| \geq \frac{1}{2}w \\[2mm] 1, |x| < \frac{1}{2}w \end{cases}$$

$L(x)$ is the luminance of a pixel at a distance $x$ perpendicular to the center of the Ricker stripe and $w$ is the width of the stripe given by the distance of the zero-crossings (i.e., the width of the central white band). The surround factor $s$ introduced in our modified definition of the Ricker wavelet only multiplies the negative sidebands of the stripe and affects its integral, which equals zero only if $s = 1$. Again, this wavelet will take on the maximum allowed luminance of +1 in the center. We here applied $w = 5$ pixels and $s = 2.5$ if not otherwise specified. If the surround factor $s$ would lead to some stimulus pixels attaining values $L < -1$, they were clipped at −1. When demonstrating the principle of the method (Fig 1), the surround factor $s$ was set to unity or, in the case without suppressive sidebands, to zero.

Measurements with Ricker stripes were performed at 36 (if not otherwise specified) equally distributed angles from 0˚ (inclusive) to 180˚ (exclusive). For each angle, a default of 60 equally spaced stripe positions were probed with a shift of 2/3 of a pixel between two positions. Consequently, 2160 combinations of stripe angle and position were tested, with each combination being measured only once, and the resulting responses were composed into a sinogram. For measurements using a spiking process, a Gaussian smoothing of the sinogram was implemented. If not specified otherwise, the applied 2D Gaussian had a standard deviation of $\sigma_{\text{pos}} =$ 2.5% of the size of the simulated area (corresponding to a standard deviation of one pixel) in the spatial dimension and $\sigma_{\text{ang}} = 5$˚ in the angle dimension. The resulting sinograms were then reconstructed using FBP with a ramp filter. Note that the resolution of the reconstruction, determined by the distance between stripe positions, is slightly higher than the resolution of the simulated area.

## Calculation of F-scores

To quantify the quality of a reconstruction obtained by FBP, we first identified all local maxima in the reconstruction. We defined a local maximum as a pixel with a value at least as large as any of its eight neighbors. Next, we discarded any local maximum smaller than 30% of the global maximum of the reconstruction. We also discarded any local maximum that lay outside a circle centered on the reconstruction area with a diameter of 90% the reconstruction area's extent. We considered all remaining local maxima to be the detected hotspots.

We then identified matches between model subunits and detected hotspots. A hotspot was considered to correspond to a subunit if it lay within the 0.75 σ ellipse of that subunit. The 0.75 σ ellipses never overlapped, and if two hotspots lay within one 0.75 σ ellipse of the same subunit, only one hotspot was judged to correspond to the subunit while the other was not considered a match. Consequently, this procedure gave us a number of true positives (hotspots corresponding to subunits), false positives (hotspots not corresponding to a subunit), and false negatives (subunits not detected by any hotspot). As a measure of reconstruction quality, we then calculated the F-score defined as the harmonic mean of precision (true positives over number of hotspots) and sensitivity (true positives over number of subunits):

$$F = \frac{2}{\frac{1}{\text{precision}} + \frac{1}{\text{sensitivity}}} = \frac{2 \cdot \text{true positives}}{2 \cdot \text{true positives} + \text{false positives} + \text{false negatives}}$$

Since the F-score of the reconstruction varies from subunit layout to subunit layout, any numbers given in the text or figures were calculated as averages over 1000 different layout instantiations, resulting in a standard error of the mean below 0.01 in all cases. All F-scores given in text and figures concern models employing a spiking process. Simulations were performed using Python and ran on a desktop computer.

## Experimental procedures

For the experimental test of our method, we used the retina of a 12-year-old adult male marmoset monkey (*Callithrix jacchus*). The retinal tissue was collected directly after the animal was killed for use by other researchers. Following the enucleation procedure, the eyes underwent dissection, during which the cornea, lens, and vitreous humor were extracted to obtain access to the retinal tissue. The retina was then placed in a light-tight chamber that contained Ames' medium (Sigma-Aldrich, Munich, Germany), supplemented with 4 mM D-glucose and oxygenated with a mixture of 95% $O_2$ and 5% $CO_2$. To maintain a pH of 7.4, the medium was buffered with 20 mM $NaHCO_3$. Following a dark-adaptation period of several hours, during which retinal pieces for other experiments were prepared, a piece of peripheral retina was excised, isolated from the pigment epithelium and placed on a multielectrode array (Multi-Channel Systems, Reutlingen, Germany) with 252 electrodes spaced 60 μm apart and sized 10 μm, which had been coated with poly-D-lysine. The entire preparation process was carried out under infrared illumination using a stereomicroscope equipped with night-vision goggles.

While recording from the retina piece, it was continuously supplied with the oxygenated Ames' medium at a flow rate of 8–9 ml/min. To maintain a stable temperature around 33˚C, an inline heater (PH01, MultiChannel Systems, Reutlingen, Germany) and a heating element beneath the array were employed. The recorded multielectrode array signals were amplified and band-pass filtered to a frequency range of 300 Hz to 5 kHz, and saved to disk using the software MC-Rack 4.6.2 (MultiChannel Systems) at a sampling rate of 25 kHz. To identify and sort spikes from the recordings, a modified version of Kilosort [92] was used. The original version can be accessed at https://github.com/MouseLand/Kilosort, while the modified version

can be found at https://github.com/dimokaramanlis/KiloSortMEA. The output generated by Kilosort was manually reviewed and curated using the software Phy2 (https://github.com/cortex-lab/phy), discarding all units without a distinct cluster of voltage traces or without a clear refractory period.

Custom-made software coded in C++ and OpenGL was utilized to generate visual stimuli. The stimuli were monochromatic white and displayed on a gamma-corrected RGB-OLED monitor (eMagin) with an 800-by-600 pixel resolution and a refresh rate of 85 Hz. When projected onto the retina through a telecentric lens (Edmund Optics), the pixel size on the retina was 7.5 μm by 7.5 μm. The stimuli described in this manuscript and the gray background illumination between stimuli had a mean irradiance of 5.45 mW/m$^2$. We calculated the isomerization rates of the photoreceptors following the formula in [93] with literature values for peak sensitivities and collecting areas for marmoset and macaque monkeys [94–97]. This led to 450 isomerizations per photoreceptor per second for S-cones, 2900 for M-cones, and 8600 for rods, indicating a low photopic light regime. The projection of the stimulus screen was focused on the photoreceptors before the start of the experiment, which was confirmed by monitoring through a microscope.

### Basic characterizations of recorded cells

To demonstrate the spatially nonlinear integration of certain ganglion cells, we displayed a reversing-grating stimulus. Square-wave gratings (100% Michelson contrast) with bar widths of 7.5 μm, 15 μm, 30 μm, 60 μm, 120 μm, 240 μm, 480 μm, and 6000 μm (full-field) and correspondingly 1, 1, 2, 2, 4, 4, 8, and 1 different spatial phases were presented for 12.5 s each (preceded by 1 s of full-field gray at mean intensity), reversing in contrast every about 0.5 s. The entire sequence was repeated once for a total stimulus duration of about 10 minutes. We calculated PSTHs with a duration of two reversals and a bin size of 10 ms for visualization.

We estimated the receptive fields of cells by computing the spike-triggered average (STA) from responses to a spatiotemporal binary white noise stimulus on a checkerboard grid [1,19]. Each stimulus field had a size of 15 μm by 15 μm and was randomly updated every four frames (i.e., 47 ms) to either black or white with 100% Michelson contrast. The stimulus alternated between distinct sequences of 3825 white noise images (3 min) and a fixed white noise sequence of 652 images (~31 s) for a total time of about one hour. In this study, we only used the non-fixed white noise. The STA was obtained with a temporal window of 42 frames (~0.5 s) at single-frame resolution. From the spatiotemporal STA, smoothed with a spatial Gaussian filter of 60 μm standard deviation, we then detected the element that had the largest absolute value. We defined the temporal component of the STA as the unsmoothed time course of that pixel and the spatial component as the unsmoothed STA frame of that element. We normalized the temporal component to a Euclidean norm of unity, fitted a 2D Gaussian to the spatial component, and calculated the effective receptive field diameter as the diameter of a circle with the same area as the 1.5 σ ellipse of the Gaussian.

To estimate a ganglion cell's output nonlinearity, we constructed an LN model, using the spatial and temporal STA components as filters. To enhance signal quality and reduce noise, we truncated the length of the temporal filter to 0.25 seconds. Additionally, only pixels falling within the smallest rectangular window that still contained the 3 σ ellipse of the fitted Gaussian were considered in this computation. Both temporal and spatial filter were normalized to unit Euclidean norm. Applying the temporal and spatial filters to the stimulus yielded a generator signal for each frame of the white noise stimulus. We grouped the generator signals into ten bins with the same number of data points. The average generator signal of each bin in

conjunction with the average of the corresponding spike count gave an estimation of the cell's contrast-response relationship.

We also computed a 50 ms long autocorrelation function for a cell's spike train from the non-fixed parts of the white noise stimulus with a resolution of 0.04 ms, smoothed with a Gaussian of 0.4 ms standard deviation, and normalized to a sum of unity.

We manually identified 31 Off and 18 On parasol cells in the dataset, based on their fast biphasic filters, effective receptive field diameters in the expected range of roughly 100 μm to 150 μm [48], and tiling of visual space by their receptive fields.

## Stimulus and analysis for subunit identification in experiments

For the application in experiments, the tomographic stimulus was slightly modified in some aspects compared to the analyses of model simulations. While the Ricker stripe profile remained unchanged, we flipped it in polarity so that the center of the stripe was at maximum black (-100% Weber contrast). A surround factor of $s = 1.5$ was applied to the white sidebands, and the width of the Ricker stripes, i.e., the distance of the zero-crossings, was chosen as $w = 45$ μm. Furthermore, in contrast to the simulations, we displayed multiple parallel stripes simultaneously across the entire stimulation area with a center-to-center distance of 375 μm. Stripes were flashed for 153 ms (13 frames) separated by 447 ms (38 frames) of full-field gray at background intensity. The stripe angles ranged from 0˚ to 175˚ in steps of 5˚. For each angle, 75 equally spaced shifts of position, ranging from 0 μm to 370 μm with a step size of 5 μm, were applied to the stripes (perpendicular to their orientation), so that the maximum shift was one step less than the stripe distance. Each combination of stripe angle and stripe position was flashed once in randomized order, before the stimulus was repeated in a new random order. In total, the tomographic stimulus was presented for almost 1.5 hours with each combination presented at least three times.

For each presented combination of stripe angle and position, we determined the average number of spikes elicited between stimulus onset and offset to compose a sinogram. We corrected the sinograms for the positioning of the receptive field by shifting the values in each sinogram row, i.e., for each angle, such that the value at the center corresponded to the stripe position with the smallest distance to the center of the receptive field. Here, we used the position of the 2D Gaussian fitted to the spatial STA from the white noise analysis as an estimate of the receptive field position. Next, we applied a Gaussian smoothing with a standard deviation of $\sigma_{pos} = 7.5$ μm and $\sigma_{ang} = 5$˚. The processed sinograms were then used for reconstruction of subunit layouts by applying the same FBP method used for the simulations.

We also analyzed the responses to the offsets of stripes, where we included all spikes during the 153 ms (i.e., the same duration as for the onset) after the offset. Sinogram and FBP analyses were performed analogously. Furthermore, we analyzed the first and second half of the recording separately, by splitting the measurement into the first roughly 45 minutes and second 45 minutes and analyzing both independently.

## Acknowledgments

We thank Fred Rieke for advice on experiments with the primate retina.

## Author Contributions

**Conceptualization:** Steffen Krüppel, Tim Gollisch.

**Data curation:** Steffen Krüppel.

**Formal analysis:** Steffen Krüppel.

**Funding acquisition:** Steffen Krüppel, Tim Gollisch.

**Investigation:** Steffen Krüppel, Mohammad H. Khani, Helene M. Schreyer, Shashwat Sridhar, Varsha Ramakrishna, Sören J. Zapp, Dimokratis Karamanlis.

**Methodology:** Steffen Krüppel, Mohammad H. Khani, Matthias Mietsch, Dimokratis Karamanlis, Tim Gollisch.

**Project administration:** Tim Gollisch.

**Resources:** Tim Gollisch.

**Software:** Steffen Krüppel.

**Visualization:** Steffen Krüppel.

**Writing – original draft:** Steffen Krüppel.

**Writing – review & editing:** Steffen Krüppel, Tim Gollisch.

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
