## [Decision Letter · Decision Letter 0]

24 Feb 2024

Dear Tim,

Thank you very much for submitting your manuscript "Applying Super-Resolution and Tomography Concepts to Identify Receptive Field Subunits in the Retina" for consideration at PLOS Computational Biology.

As with all papers reviewed by the journal, your manuscript was reviewed by members of the editorial board and by an independent reviewer. In light of the review (below this email), we would like to invite the resubmission of a significantly-revised version that takes into account the reviewer's comments.

It is unusual that we make a decision based on only one review, but in this case 1) I don't want to further delay the process (it has been quite difficult to find reviewers), and 2) I think that the enclosed review is sufficient at this stage. That being said, it remains possible that we will ask for additional reviews for a revised manuscript.

We cannot make any decision about publication until we have seen the revised manuscript and your response to the reviewers' comments. As stated, your revised manuscript is also likely to be sent to reviewers for further evaluation.

Sincerely,

Lyle Graham

Section Editor

PLOS Computational Biology

Reviewer's Responses to Questions

**Comments to the Authors:**

Reviewer #1: This paper introduces an approach to identify receptive field subunits. Subunits form an important anatomical and computational part of how sensory neurons combine their inputs. In the retina, bipolar cells provide a major source of subunits in ganglion cell receptive fields. These subunits can cause ganglion cells to response to spatial features finer than expected from the receptive field itself, and contribute importantly to a number of interesting computations. But progress in investigating these issues has been slowed by the difficulty identifying subunits based on ganglion cell responses. Several approaches have been introduced previously, but all have substantial limitations. Hence a new approach is welcome. The current paper is clearly written and many aspects of the approach introduced are nicely documented. There are a few areas in which additional analyses would substantially improve the paper and better situate it relative to other approaches to subunit identification:

Time required

One of the arguments made in favor of the new approach is that it requires less time to identify subunits. It is not clear from the results in the paper whether this is the case, and if it is how much of a benefit the approach represents. In the Shah et al. (2020) paper, subunits are identified based on white noise stimulation in ~20 min. The approach presented here would appear to require a similar amount of time. But that is a very qualitative comparison - and one or the other approach may be more efficient. The paper would benefit substantially from a direct comparison of a noise-based approach (e.g. the non-negative matrix factorization approaches used by the Gollisch group in the past or the Shah et al. approach) and the current approach.

Overlapping subunits

Many RGC types receive input from several bipolar types (as mentioned in the Discussion). That might be expected to create overlapping subunits. The Discussion mentions that these may be hard to resolve, but it would be nice to see an analysis of this situation and a discussion based on this analysis of its importance.

Robustness to multiple sources of subunits

Work from some of the same authors shows nicely that subunits are already present in the bipolar input signals. How robust is the approach to having several sequential sources of subunits? An analysis of this possibility and how well (or not well) it is handled would enhance the paper.

Experimental validation

The experimental validation of the approach is an important addition but is less complete than the remainder of the paper. At present, it consists of a few example cells but no real analysis of cell populations. Is the number of subunits identified across ganglion cells of the same type consistent? What about the subunit sizes? The number of identified subunits (e.g. four in the case of the On parasol shown in Fig. 6) seems quite low; can you compare that to expectations from anatomy and comment on any discrepancy? On the technical side, is the 375 micron separation of the stripes sufficient to avoid surround activation?

Nature of errors

Are the errors that the algorithm makes largely missing or mislocalizing subunits (or a combination of the two)?

Extend analysis of accuracy as number of subunits increases

Prior approaches to identify subunits often end up with fewer than expected based on anatomical considerations, suggesting that the functional subunits are in fact combinations of several anatomical subunits. This has long been puzzling. Figure 3 investigates this issue nicely. It would be helpful to extend that analysis to larger numbers of subunits (16 is probably close to a minimal number for most ganglion cell types, and many likely have quite a few more subunits). More exploration of how time and number of subunits trade off would also be helpful - e.g. if the number of subunits doubled, how much additional time would be required to compensate and recover equally accurate estimates?

Responses to bar offsets

Have you tried using responses to both the onset and the offset of the bars in your analysis (similar to looking at a frequency-doubled response to isolate the nonlinear component of a contrast-reversing grating)? It might decrease sensitivity to the RF itself and increase sensitivity to the subunits.

**Have the authors made all data and (if applicable) computational code underlying the findings in their manuscript fully available?**

Reviewer #1: Yes

PLOS authors have the option to publish the peer review history of their article (what does this mean?). If published, this will include your full peer review and any attached files.

Reviewer #1: No
---

## [Decision Letter · Decision Letter 1]

28 Jul 2024

Dear TIm,

We are pleased to inform you that your manuscript 'Applying Super-Resolution and Tomography Concepts to Identify Receptive Field Subunits in the Retina' has been provisionally accepted for publication in PLOS Computational Biology.

It's not usual that we consider just a single reviewer, but in this case I think the decision was more than justified, and I'm glad for their approbation.

Best regards,

Lyle J. Graham

Section Editor

PLOS Computational Biology

Reviewer's Responses to Questions

**Comments to the Authors:**

Reviewer #1: The authors have done a very nice job revising the paper. I think the new material substantially strengthens the paper (particularly the more systematic exploration of the limitations of the approach and the somewhat surprising ability to uncover complex subunit layouts). Nice work!

**Have the authors made all data and (if applicable) computational code underlying the findings in their manuscript fully available?**

Reviewer #1: Yes

PLOS authors have the option to publish the peer review history of their article (what does this mean?). If published, this will include your full peer review and any attached files.

Reviewer #1: No

---

## [Editor Report · Acceptance letter]

16 Aug 2024

PCOMPBIOL-D-23-01940R1 

Applying Super-Resolution and Tomography Concepts to Identify Receptive Field Subunits in the Retina

Dear Dr Gollisch,

I am pleased to inform you that your manuscript has been formally accepted for publication in PLOS Computational Biology. Your manuscript is now with our production department and you will be notified of the publication date in due course.

With kind regards,

Lilla Horvath
